# Single-cell profiling coupled with lineage analysis reveals vagal and sacral neural crest contributions to the developing enteric nervous system

Jessica Jacobs-Li[†], Weiyi Tang[†], Can Li, Marianne E Bronner*

Division of Biology and Biological Engineering, California Institute of Technology, Pasadena, United States

**Abstract** During development, much of the enteric nervous system (ENS) arises from the vagal neural crest that emerges from the caudal hindbrain and colonizes the entire gastrointestinal tract. However, a second ENS contribution comes from the sacral neural crest that arises in the caudal neural tube and populates the post-umbilical gut. By coupling single-cell transcriptomics with axial-level-specific lineage tracing in avian embryos, we compared the contributions of embryonic vagal and sacral neural crest cells to the chick ENS and the associated peripheral ganglia (Nerve of Remak and pelvic plexuses). At embryonic day (E) 10, the two neural crest populations form overlapping subsets of neuronal and glia cell types. Surprisingly, the post-umbilical vagal neural crest much more closely resembles the sacral neural crest than the pre-umbilical vagal neural crest. However, some differences in cluster types were noted between vagal and sacral derived cells. Notably, RNA trajectory analysis suggests that the vagal neural crest maintains a neuronal/glial progenitor pool, whereas this cluster is depleted in the E10 sacral neural crest which instead has numerous enteric glia. The present findings reveal sacral neural crest contributions to the hindgut and associated peripheral ganglia and highlight the potential influence of the local environment and/or developmental timing in differentiation of neural crest-derived cells in the developing ENS.

**\*For correspondence:**
mbronner@caltech.edu

[†]These authors contributed equally to this work

## Editor's evaluation

This paper is useful for researchers in the field of enteric neuroscience and peripheral nervous system development. The single cell RNA-sequencing based analysis of the developing chicken ENS, demonstrates differential cell identity contribution from the sacral and vagal neural crest and influence of the local distal embryonic environment for final differentiation. A basic classification scheme of neuronal cell types in the chicken combined with analysis of a more mature embryonic stages and functional data will however be needed in the future to determine the role of differential stem cell origin for final neuronal composition in the distal gut of chicken.

## Introduction

The enteric nervous system (ENS) is the largest component of the peripheral nervous system and plays a critical role in regulating gut motility, homeostasis, and interactions with the immune system and gut microbiota (***Nagy and Goldstein, 2017***). In amniotes, the ENS consists of millions of neurons with motor, sensory, secretory, and signal transduction functions, as well as a larger number of supportive enteric glia. Interestingly, enteric glia recently have been shown to retain neurogenic potential via reentrance into a progenitor-like state (***Laddach et al., 2023***). Together these diverse neurons and glia

form a highly orchestrated network of physically and chemically connected cells embedded between the muscle and mucosal layers of the gastrointestinal system (*Fleming et al., 2020*). Due to the vast number of cells and its capacity for autonomic regulation, the ENS is often referred to as 'a second brain' (*Gershon, 1999*).

The neurons and glia of the ENS arise from the neural crest, a migratory stem cell population, emigrating from the closing neural tube. This transient population consists of four subpopulations designated from rostral to caudal along the body axis as cranial, vagal, trunk, and sacral. Best studied in mouse and chick embryos, much of the ENS is derived from 'vagal' neural crest cells that arise in the caudal hindbrain (adjacent to somites 1–7) at chick Hamburger Hamilton (HH) stage 10, approximately embryonic (E) day 1.5. These cells enter the foregut and migrate caudally to populate the entire length of the gut by E8 in the chick embryo (*Le Douarin and Teillet, 1973*), as well as giving rise to nerve-associated Schwann cell precursors (SCPs) that later invade the gut (*Uesaka et al., 2015*; *Espinosa-Medina et al., 2017*). However, there is an additional neural crest contribution to the ENS from the sacral neural crest population (*Le Douarin and Teillet, 1973*). First observed by Le Douarin and Teillet in quail-chick chimeric grafts, the sacral neural crest arises caudal to somite 28 at HH 17–18 (E2.5), migrates to the dorsal side of the developing gut, and forms the paired pelvic plexuses and Nerve of Remak at E3.5 (*Anderson et al., 2006*; *Burns and Douarin, 1998*; *Yntema and Hammond, 1955*; *Figure 1A*). The Nerve of Remak, which is specific to bird, is closely associated with the hindgut and has been described as a staging ground for many neural crest-derived cells which migrate to the gut along extrinsic axons to colonize the post-umbilical gut by E8 and rapidly expand in number by E10 (*Burns and Douarin, 1998*; *Pomeranz et al., 1991*).

Dysregulation of ENS development is responsible for enteric neuropathies such as Hirschsprung's disease which affects 1 in 5000 live births (*Amiel et al., 2008*), and is characterized by a paucity or absence of neurons in the distal colon resulting in potentially lethal obstruction and increased risk of infection (*Lake and Heuckeroth, 2013*; *Ji et al., 2021*; *Lourenção et al., 2016*). While the etiology of Hirschsprung's disease is not completely understood, it is thought that insufficient migration or proliferation of vagal neural crest precursors results in neuronal deficits, particularly in the hindgut due to the long distance needed for precursor cells to reach their destination. Grafting sacral neural crest cells in place of ablated vagal neural crest results in isolated ganglia in myenteric and submucosal plexuses. However, the grafted sacral neural crest's contribution to the ENS was insufficient to compensate for the lack of vagal-derived cells, suggesting that there may be intrinsic differences between these two populations (*Burns et al., 2000*). Molecular cues such as GDNF (*Young et al., 2001*) and ET3 (*Nagy and Goldstein, 2006*) are essential for migration and differentiation of vagal neural crest during early development and mutations in these genes are common in patients with Hirschsprung's disease (*Kenny et al., 2010*). However, it is unknown if these genes are involved in the development of the sacral neural crest.

Recent studies have proposed that lack of rostrocaudal migration of the vagal neural crest may not be the only cause of enteric neuropathies. A distinction between the ENS of the foregut/midgut versus hindgut is that the former arises solely from vagal neural crest-derived cells, whereas the latter is populated by both vagal and sacral neural crest-derived cells (*Burns and Douarin, 1998*). Thus, a complete understanding of ENS ontogeny requires more thorough characterization of possible differences between vagal neural crest contributions to the pre-umbilical versus post-umbilical gut and characterization of sacral neural crest-derived contributions to the hindgut and associated peripheral ganglia. Open questions include: What cell types are derived from the sacral neural crest? Is the sacral neural crest population distinct from the vagal, or do they have shared derivatives? Does the post-umbilical gut possess special cell types absent in the pre-umbilical region? Defining the transcriptional landscape of sacral and vagal neural crest-derived cells along the entire length of the gut holds the promise of revealing similarities and differences between these populations.

To tackle these questions, we combined single-cell transcriptomics with a recently developed lineage tracing method in which replication-incompetent avian (RIA) retroviruses can be used to infect specific axial levels of the neural tube of the developing chick embryos to permanently express an inherited fluorophore (*Tang et al., 2019*). By infecting either vagal or sacral neural crest populations, RIA retroviral infection permits region-specific lineage tracing without the need for transplantation or Cre-mediated recombination that can result in ectopic expression. This enables transcriptional profiling of vagal- or sacral-derived RIA-labeled ENS cells in the pre- and post-umbilical gut at

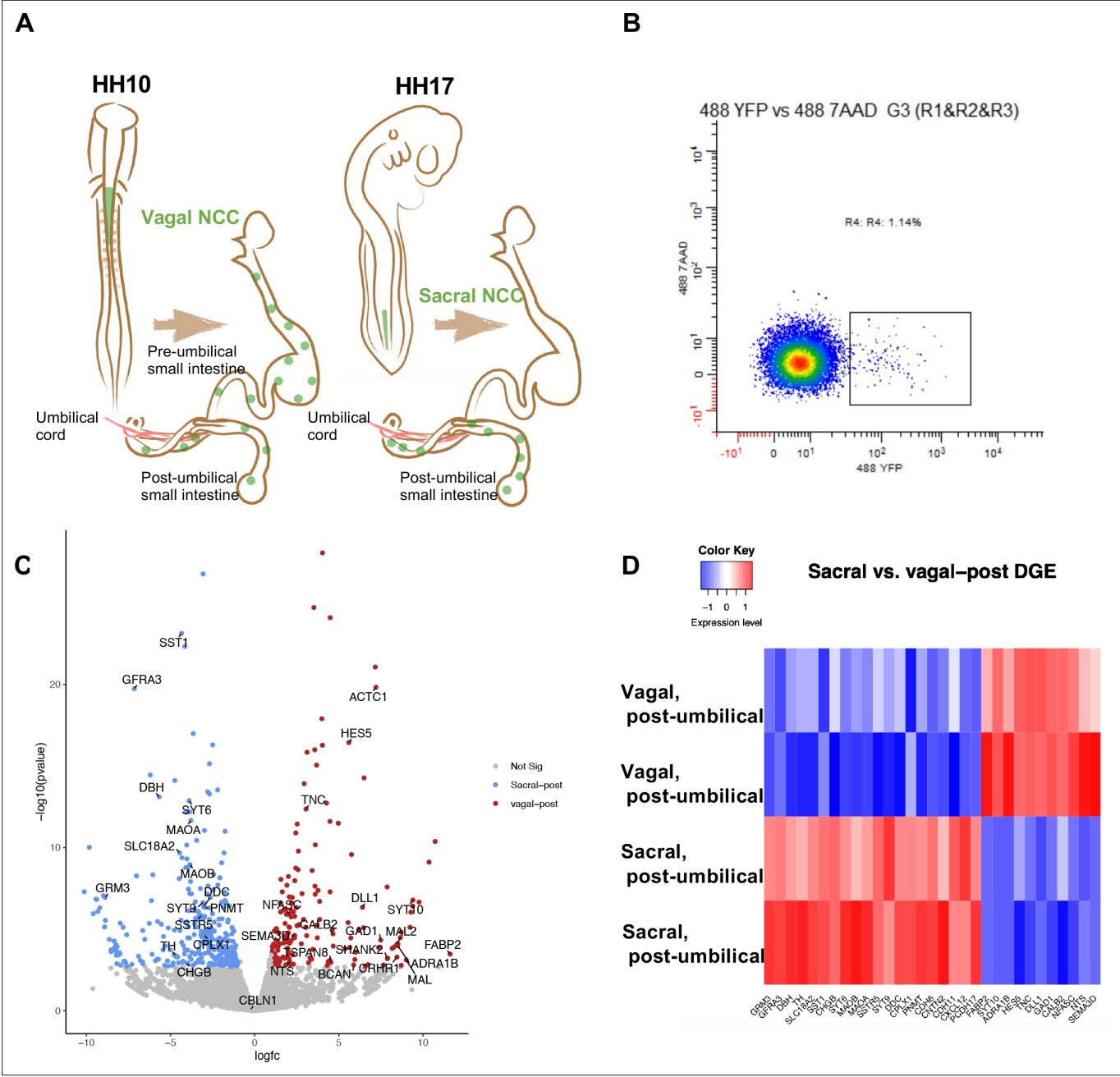

**Figure 1.** Bulk RNA-seq of vagal and sacral neural crest derived cells in the post-umbilical enteric nervous system (ENS). (**A**) Schematic diagram describing experimental procedure for viral labeling. Vagal and sacral neural crest cells were labeled by H2B-YFP (green) in separate embryos. The post-umbilical gastrointestinal tracts, including accompanying ganglia, were dissected at E10 for dissociation. (**B**) YFP+ cells from the post-umbilical region derived from vagal or sacral neural crest (NC) were sorted via FACS. (**C**) Volcano plot describing differentially expressed genes of sacral (sacral-post, blue) and vagal neural crest cells in the post-umbilical gut (vagal-post, red). Genes with fold change greater than 2 and p-value<0.05 are colored. (**D**) Heatmap highlighting selected genes related to neuronal functions from differential gene expression analysis in sacral and vagal-post ENS populations (with two replicates per condition). Genes are ordered based on significance level and fold change.

The online version of this article includes the following figure supplement(s) for figure 1:

**Figure supplement 1.** DiI-labeling of the sacral neural crest.

**Figure supplement 2.** Diagram of dissected tissue for single-cell RNA-sequencing.

single-cell resolution. To characterize these cell populations, we chose E10 (similar to E16 mouse and 8 wk post-conception in human) as a starting time point. By E10, the vagal neural crest is undergoing the process of differentiation along the entirety of the gut while in the post-umbilical gut the sacral neural crest has formed the Nerve of Remak and has begun neurogenesis. Thus, this time point reflects a stage in which cells from each population contain both precursors and some differentiated neuronal subtypes.

Our results reveal both interesting similarities and differences between pre-umbilical vagal, post-umbilical vagal, and sacral neural crest-derived cells. While the vagal neural crest is the only contributor to *CALB2/TAC1/PBX3+* neurons in both the pre- and post-umbilical gut at E10, the sacral neural crest contributes over 50% of cells to neuronal subtypes that are unique to the post-umbilical gut. Our in vivo analysis at E10 reveals sacral-derived neurons in the submucosal and myenteric plexuses as well as a major population residing within the Nerve of Remak. Interestingly, the vagal neural crest population in the post-umbilical gut shares more clusters with the sacral neural crest than the pre-umbilical vagal neural crest. The exception is that the sacral-derived population at this time point appears to be depleted in a neuronal/glial precursor present in the vagal-derived population. Trajectory analysis suggests that many sacral neural crest-derived cells are predicted to be enteric glia/SCPs, which have been shown to maintain a neurogenic potential in vitro and in an injury model (*Laddach et al., 2023*). Collectively, the data provide a transcriptomic reference for the developing chick ENS and associated peripheral nerves, expanding our understanding of the role of the sacral neural crest, a largely understudied stem cell population. Our results further suggest that the hindgut environment and/or developmental timing may influence cell fate decisions in the ENS.

## Results

### Sacral and vagal neural crest exhibit distinct transcriptional profiles at the population level

As a first step in assessing contributions of vagal and sacral neural crest in the post-umbilical region, we selectively labeled either the vagal or sacral neural crest population using a novel RIA retrovirus lineage-tracing technique (*Tang et al., 2019*; *Tang et al., 2021*) for bulk RNA-seq analysis. To this end, the neural tube at the level of the caudal hindbrain was injected with RIA retrovirus carrying a YFP expression cassette at HH10 (~E1.5) to label vagal neural crest cells, or below the level of somite 28 at HH17 (E2.5) to label the sacral neural crest (*Figure 1A*).

Injection of lineage tracer into the neural tube of chick embryos is a frequently used and accurate method to label the premigratory neural crest (*Nakamura and Funahashi, 2001*; *Serbedzija et al., 1991*). RIA virus has been shown to specifically label the vagal neural crest (*Tang et al., 2019*; *Tang et al., 2021*), but has not been previously applied to the sacral level. As it takes ~2 d for YFP expression mediated by RIA infection to become detectable, we turned to an alternative lineage label, the lipophilic dye DiI, to demonstrate the specificity of labeling the sacral neural tube/neural crest since it is immediately visible upon injection. DiI was injected in the same manner as RIA into the sacral neural tube of HH17 (E2.5) chick embryos. Shortly after injection, we observed specific labeling confined to the lumen of the neural tube with no labeling of adjacent tissue; subsequently, DiI was observed in the neural crest migratory streams 48 hr after injection (~HH24-25, E5) (*Figure 1—figure supplement 1*).

Next, we used the RIA virus encoding YFP to obtain pure populations of either vagal or sacral neural crest cells from the E10 post-umbilical gut, including the closely associated Nerve of Remak and pelvic plexuses (*Figure 1A*, *Figure 1—figure supplement 2*). The E10 time point reflects a stage at which both vagal and sacral neural crest cells have populated the post-umbilical gut for 2 d, enabling capturing of both precursors and some differentiated neuronal subtypes. After dissociation, YFP+ cells were sorted using FACS (*Figure 1B*). Similar regions from three guts were pooled as a replicate, with each library containing 2000 cells.

Differential gene expression analysis revealed intriguing distinctions between vagal and sacral neural crest cells in the post-umbilical gut at the population level. Genes enriched in the sacral population include *SST1/SSTR, DBH, TH, DDC, PNMT,* and *SLC18A2. GRM3* expression is more abundant in the sacral population, which may reflect a transient state of differentiation. In addition, we observed upregulation of *GFRA3*, the receptor for artemin and *CXCL12* which is related to signaling during cell migration (*Figure 1C and D*). Conversely, the vagal post-umbilical population expresses the adrenergic

receptor *ADRA1B*, enzyme *GAD1, CALB2,* and *NTS*. Additionally, the population expresses genes related to neural crest and neuronal migration such as *SEMA3D* and *TNC*. *HES5* and *DLL1* expression indicate functions for Notch/Delta signaling in the vagal-post-umbilical population (*Figure 1C and D*).

## Single-cell transcriptome profiling of the chick ENS

Whereas portions of the mammalian including the human ENS have been transcriptionally profiled at the single-cell level (*Nakamura and Funahashi, 2001*; *Serbedzija et al., 1991*), there is much less information about the developing chick ENS, and particularly the sacral-derived subpopulation. To understand the transcriptional profile of vagal and sacral neural crest-derived cells at single-cell resolution, we performed viral labeling as described above (*Figure 1A*) and collected three distinct neural crest populations at E10: vagal neural crest from the pre-umbilical gut (vagal-pre; three guts pooled per replicate), vagal neural crest from the post-umbilical gut and associated peripheral ganglia (including the Nerve of Remak) (vagal-post; six guts pooled per replicate), and sacral neural crest from the post-umbilical region plus associated peripheral ganglia (sacral; six guts pooled per replicate). After FACS isolation of YFP+ cells, 4.6k–5k cells were sequenced for each replicate producing transcriptomic profiles for cells. In total, 26,993 of these cells (3.2k–5.4k cells per replicate) possessed RIA viral genome transcripts (two or more), indicating true infection, and were used to generate a single-cell profile that organized into 15 clusters (*Figure 2A*; *Zheng et al., 2017*; *Bray et al., 2016*; *Butler et al., 2018*). To ascribe cluster identity, we first performed gene expression heatmap analysis for the top 10 gene markers in clusters 0–14 (C0–C14), displaying the most upregulated genes with focus on genes relevant to neuronal, progenitor, glial, and non-neural identity (*Figure 2B*). UMAPs were generated for pre-umbilical vagal, post-umbilical vagal, and post-umbilical sacral (*Figure 2C*). We also calculated the proportion of cells that each population contributes to the clusters (*Figure 2—figure supplement 1*).

Gene markers were analyzed for each cluster (*Supplementary file 1*) and violin plots were generated to observe differences in gene expression profiles between clusters (*Figure 3B*). Based on known cell-type markers, clusters were preliminarily classified as neuronal, glial, progenitor, or non-neural (*Figure 3A*). Neuronal clusters (C2, C4) had high expression of *ELAVL4* (*Figure 3A and B*), a marker of early post-mitotic neurons (*Akamatsu et al., 2005*). Both C2 and C4 express the neuropeptide *GAL*, cholinergic receptor *CHRNA3*, and tyrosine signaling kinase *RET*. C2 expresses the neuropeptide *VIP* and tachykinin (*TAC1*) and is predominantly vagal-derived (post-umbilical vagal – 81%; pre-umbilical vagal – 16%) (*Figure 2C*, *Figure 2—figure supplement 1*). In contrast, C4 expresses *DDC*, *PNMT*, *CHGB* and is specific to the post-umbilical gut (vagal –35%; sacral – 65%).

We identified enteric glial clusters (C1, C6, C8) based on expression of the neural crest marker *SOX10* (*Kim et al., 2003*) and enteric glial markers *PLP1* (*Rao et al., 2015*), *PMP22* (*Hagedorn et al., 1999*), *S100A10* (*Ferri et al., 1982*), *FRZB*, and *ZEB2* (*Hegarty et al., 2015*; *Figure 3A and B*). Within this enteric glial group, there is heterogeneity in expression of glial genes, including SCP genes. Both C1 and C8 express the gene *CDH19,* while only C8 expresses the canonical SCP marker *MPZ* (*Jessen and Mirsky, 2019*). C6's expresses *HEYL*, *FRZB*, and *ZEB2*. These clusters are mainly comprised of post-umbilical gut populations. C1 and C8 have high proportions of sacral cells (60%, 59%) versus post-umbilical vagal (35%, 39%) and minimal contribution from the pre-umbilical vagal (5%, 2%). Likewise, C6 has low contribution from the pre-umbilical vagal but has larger contribution of vagal cells compared to sacral in the post-umbilical gut (61% vs 39%).

To highlight enriched signaling pathways and transcription factors, we generated tables of genes implicated in signaling (signal transduction [GO:0007165], potential cell–cell signaling [GO:0007267], transcription regulation [DNA-binding transcription factors; GO:0003700], or transcription factor binding [GO:0008134]) (*Supplementary file 2*). All three clusters expressed genes associated with signaling pathways such as Wnt (*FRZB*, *SFRP2/5*, *NKD1*), Vegf (*ETS1*), Tgfb (*JUN*, *TGFB1*), and Notch (*JAG*), GABA (*GNG5*).

C0, C3, and C7 were classified as progenitors due to expression of *SOX10* and low expression of the neuronal and glial fate markers *ELAVL4* and *PLP1* (*Figure 3A and B*). C0 has high expression of *EDNRB* (*Liu et al., 2019*) and *SOX8* (*Cheung and Briscoe, 2003*), and low *ASCL1* (*Castro et al., 2011*). This putative neuroblast cluster is vagal-derived with the largest population present in the pre-umbilical gut (92%) versus post-umbilical (7%) gut (*Figure 2C*, *Figure 2—figure supplement 1*). C7 is similar but coupled with higher expression of *ASCL1*. Interestingly, both C7 and C0 are depleted

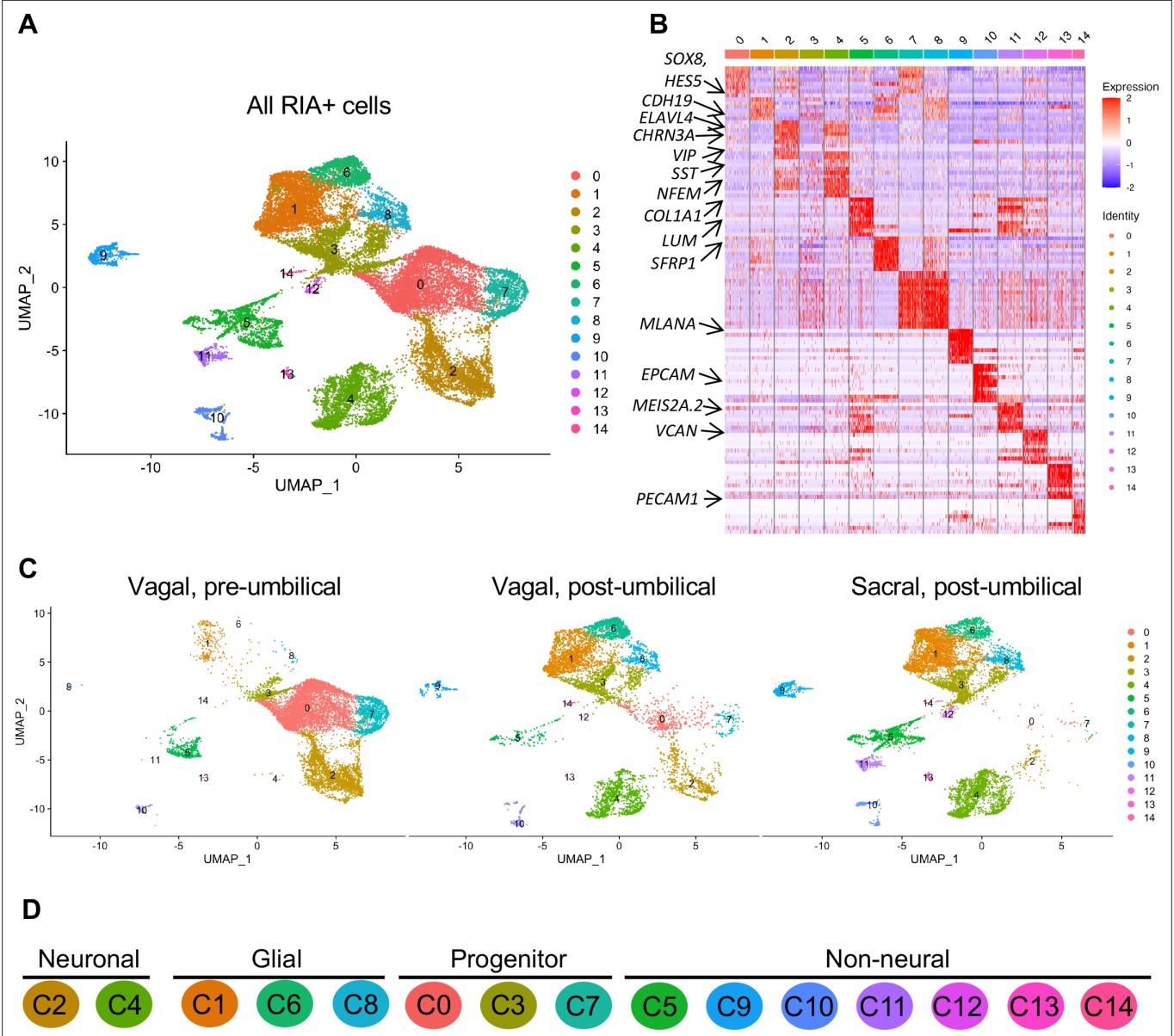

**Figure 2.** Single-cell RNA-seq of vagal and sacral-contributions to the enteric nervous system (ENS) and associated peripheral ganglia. (**A**) Uniform manifold approximation and projection (UMAP, resolution 0.3) representation of all RIA+ cells (>2 RIA transcripts; 26,993 cells) collected from the vagal and sacral-labeled embryos in both post-umbilical (including the Nerve of Remak and pelvic plexus) and pre-umbilical gastrointestinal tracts. (**B**) Expression heatmap for top 10 gene markers in clusters 0–14 (subsampled) with arrows pointing to marker genes for neural crest progenitor and glial (*SOX8, HES5, CDH19, SFRP1*), neuronal (*ELAVL4, CHRN3A, VIP, SST, NFEM*), fibroblast (*COL1A1, LUM*), melanocyte (*MLANA*), epithelial (*EPCAM*), vascular muscle (*MEIS2A.2, VCAN*), and endothelial (*PECAM1*). (**C**) UMAP representation (resolution 0.3) of each population: vagal-derived cells in the pre-umbilical gut, vagal-derived cells in the post-umbilical gut, and sacral-derived cells in the post-umbilical gut. (**D**) Key for putative cluster identities. See *Figure 3A* for greater detail.

The online version of this article includes the following figure supplement(s) for figure 2:

**Figure supplement 1.** Proportion of pre-umbilical vagal, post-umbilical vagal, and sacral populations across clusters.

in the sacral neural crest (*Figure 2C*, *Figure 2—figure supplement 1*). C3 has no clear gene profile (*Figure 2B*) and lacks differentiation markers (*Figure 3B*), potentially indicating a stem cell progenitor identity, supported by expression of the neural crest gene *LMO4* (*Ochoa et al., 2012*) and stem cell genes *LAMA4* and *LAMB1* (*Ulloa-Montoya et al., 2007*; *Figure 3B*). Sacral neural crest contributes 52% of cells in C3 compared with pre-umbilical (15%) and post-umbilical vagal (33%) (*Figure 2C*, *Figure 2—figure supplement 1*).

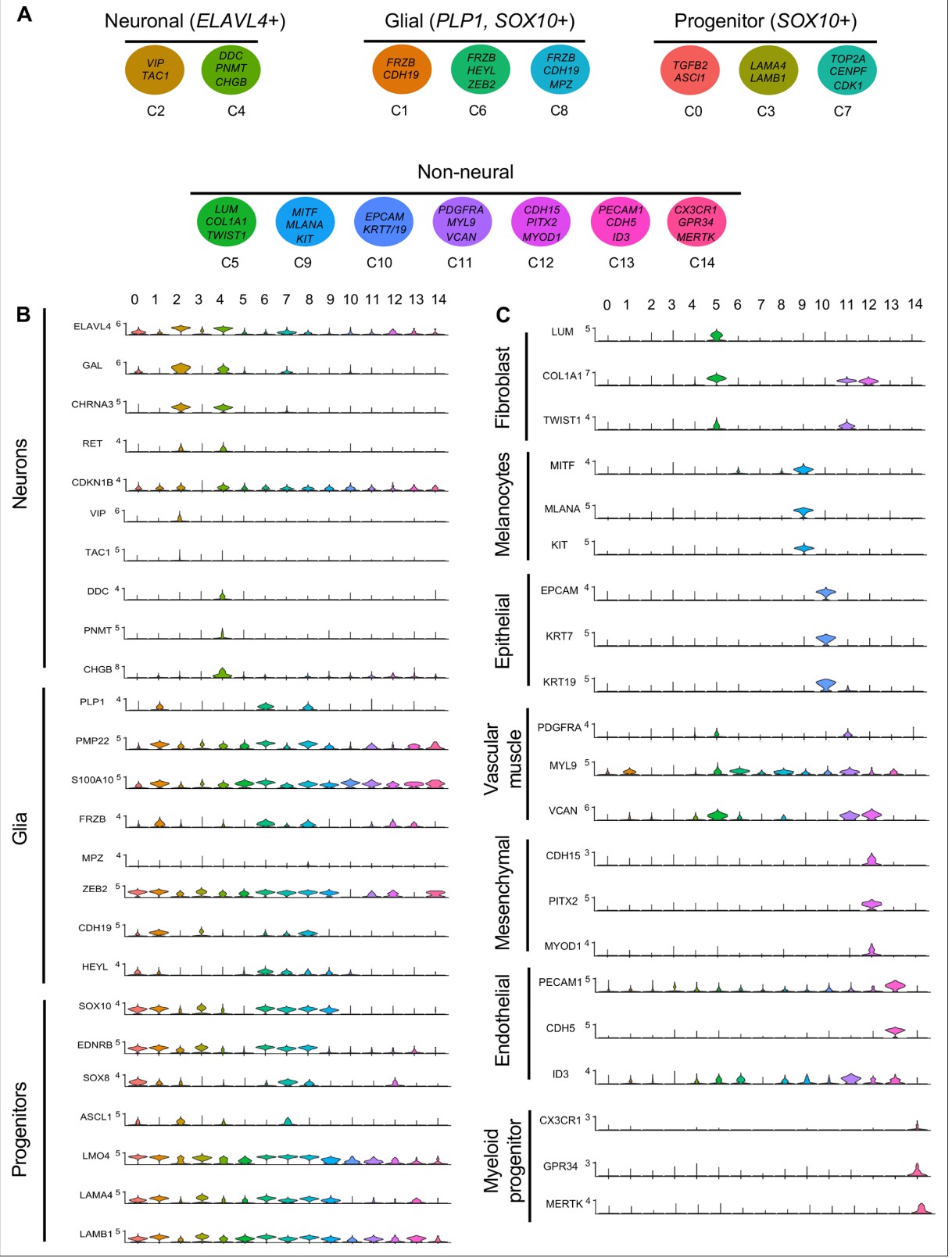

**Figure 3.** Gene expression analysis of markers associated different enteric nervous system (ENS) cell types. (**A**) Schematic diagram demonstrating genes associated with particular clusters of neuronal (*ELAVL4+*), glial (*PLP1+*, *SOX10+*), progenitor (*SOX10+*), and non-neural cell identities. (**B**) Violin plot of key genes reflecting neuronal/glial/progenitor cell fates. (**C**) Violin plot of key genes for non-neural cell identities (fibroblast, melanocyte, epithelial, vascular muscle, mesenchymal, endothelial, and myeloid progenitor).

Several small clusters C5/C9–14 contain profiles not typically associated with the ENS (*Figure 3A*). While these may be contaminating cells captured due to autofluorescence, we cannot exclude the possibility that some may be neural crest-derived since these clusters all express RIA transcript. C5/C9–C14 are likely non-neural cells due to the absence of clear neuronal or glial markers (*Figure 3B*). C5 (60% sacral, 8% post-umbilical vagal, 32% pre-umbilical vagal) (*Figure 2C*, *Figure 2—figure supplement 1*) expresses the fibroblast genes *LUM*, *COL1A1* (*Muhl et al., 2020*), and *TWIST1* (*Vincentz et al., 2013*; *García-Palmero et al., 2016*; *Figure 3C*). C9, present only in the post-umbilical gut (*Figure 2C*), has a large sacral contribution (80%) with high expression of the melanocyte genes *MITF* (*Levy et al., 2006*) (alternative name *CMI9*; *Mochii et al., 1998*), *MLANA* (*Chen et al., 2021*), and *KIT* (*Wehrle-Haller, 2003*). C10 (40% sacral, 40% post-umbilical vagal, 20% pre-umbilical vagal) expresses the epithelial genes *EPCAM* (*Trzpis et al., 2007*) and the keratin genes *KRT7/1947* (*Athwal et al., 2019* ) cells (*Figure 2C*, *Figure 2—figure supplement 1*). C11 expresses vascular muscle genes *PDGFRA*, *MYL9*, and *VCAN* (*Sorokin et al., 2020*). C12 has high expression of the mesenchymal genes *CDH15*, *MYOD* (*Kasprzycka et al., 2019*), and *PITX2* (*Gage et al., 2014*). C13 expresses *PECAM1* (*Watt et al., 1995*), *CDH5* (*Sauteur et al., 2014*), and *ID3* (*Das et al., 2015*; *Gadomski et al., 2020*), potentially indicating an endothelial cell type. C11, C12, and C13 are almost exclusively derived from the sacral neural crest (99, 97, and 95%, respectively) (*Figure 2C*, *Figure 2—figure supplement 1*). C14 expresses the macrophage genes *CX3CR1*, *GPR34*, *MERTK* (*Verheijden et al., 2015*) and is found in both sections of the gut, with greatest contribution from the post-umbilical vagal (42%) and sacral (56%) neural crest (*Figure 2—figure supplement 1*).

## Validation of marker expression by vagal versus sacral neural crest using dual retroviral lineage tracing

We next sought to validate the gene expression differences identified by single-cell RNA-seq between sacral and vagal neural crest contributions to the gut in differentially labeled cell populations. To this end, we utilized axial-level-specific retroviral labeling to sequentially mark vagal or sacral neural crest cells with different fluorophores in the same embryo. For identifying the vagal neural crest, RIA retrovirus expressing nuclear H2B-RFP was injected into the neural tube adjacent to somite 1–7 at HH10 (~E1.5). Embryos were then allowed to develop until HH17 (E2.5), at which time RIA retrovirus carrying H2B-YFP was injected into the neural tube posterior to somite 28 to label the sacral neural crest. The entire length of the gut and associated Nerve of Remak was dissected and removed for immunohistochemistry at E10 and stained with antibodies to gene products identified as differentially expressed in our scRNA-seq dataset (*Figure 3A*).

The results show that the pre-umbilical gut contained only H2B-RFP+ (yellow in figure), suggesting that only vagal but not sacral neural crest cells contributed to this region (*Figure 4—figure supplement 1*). This is consistent with previous studies using quail-chick chimerae demonstrating the absence of sacral-derived cells in the pre-umbilical gut (*Le Douarin and Teillet, 1973*). Immunohistochemistry revealed vagal RIA retrovirus-labeled cells (yellow) that co-expressed acetylcholine receptor (ACHR) (*Figure 4A–A'*, *Figure 4—figure supplement 1A–A""*) and HUC/D (ELAV) in the pre-umbilical (*Figure 4B–B'*, *Figure 4—figure supplement 1B–B""*), consistent with differentiated neurons. Additionally, there were sparsely distributed neurons marked by TH (*Figure 4C–C'*, *Figure 4—figure supplement 1C–C""*) and DBH (*Figure 4D–D'*, *Figure 4—figure supplement 1D–D""*). H2B-RFP (yellow) and P0 double-positive Schwann cells were present along the pre-umbilical region (*Figure 4E–E'*, *Figure 4—figure supplement 1E–E""*), as well as enteric progenitors or glial cells as determined by SOX10 expression (*Figure 4F–F'*, *Figure 4—figure supplement 1F–F""*).

In contrast to the pre-umbilical gut, the post-umbilical ENS contained both H2B-RFP+ cells (yellow) and H2B-YFP+ (cyan) cells, indicating a collective contribution from sacral neural crest (cyan) and vagal neural crest (yellow) (*Figure 1A*, *Figure 5*). Consistent with our scRNA-seq data, neural crest cells from different axial origins appeared to express different markers. ACHR+ cells were more abundant in the vagal-derived population throughout the post-umbilical region (*Figure 5A and A'*, *Figure 5—figure supplement 1A–A""*), whereas ACHR+ cells from sacral neural crest cells were observed primarily in the myenteric plexus of the post-umbilical gut (*Figure 5A and A"*, *Figure 5—figure supplement 2A–A""*). Both vagal (*Figure 5B and B'*, *Figure 5—figure supplement 1B–B""*) and sacral (*Figure 5B and B"*, *Figure 5—figure supplement 2B–B""*) neural crest populations differentiated into HUC/D+neurons, with the vagal-derived HUC/D+ cells residing within the myenteric and submucosal

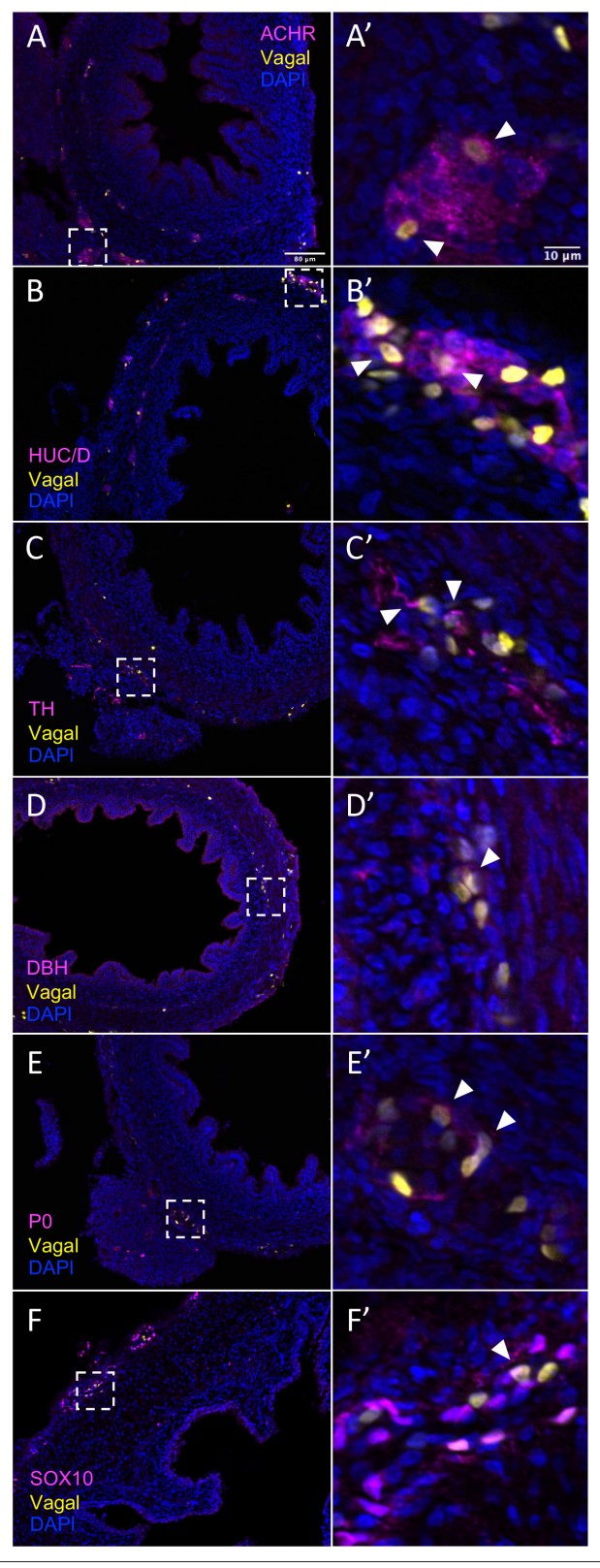

**Figure 4.** In vivo validation of vagal neural crest contributions to neuronal, glial, and progenitor cells in the pre-umbilical gut. Transverse sections through the E10 preumbilical gut reveal (**A**) acetylcholine receptor expression in some vagal neural crest-derived cells along the pre-umbilical small intestine. (**B**) Vagal neural crest also gave rise to HUC/D+ neurons in this region. (**C**) A small number of neurons expressing TH were observed sparsely distributed

*Figure 4 continued on next page*

*Figure 4 continued*

in the myenteric plexus of the pre-umbilical intestine. (**D**) DBH expressing vagally derived cells were also observed in both the myenteric and submucosal plexus of the pre-umbilical intestine. (**E**) Vagal-derived glial cells (P0+) were present in the pre-umbilical intestine. (**F**) Enteric progenitor or glial cells expressing nuclear SOX10 were present in the pre-umbilical gastrointestinal tract. Insets (**A′–F′**) show magnified regions in the corresponding dashed box. Sacral neural crest cells were absent from the pre-umbilical gut. White arrows indicate double-positive cells. Scale bars for main figure (**A–F**): 80 μm. Scale bars for insets (**A′–F′**): 10 μm.

The online version of this article includes the following figure supplement(s) for figure 4:

**Figure supplement 1.** Absence of sacral-derived cells in the E10 pre-umbilical gut.

**Figure supplement 2.** Vagal contribution to the pre-umbilical gut separated by channel.

---

plexus. While the majority of sacral-derived cells were in the Nerve of Remak, others were observed in the submucosal and myenteric plexuses (*Figure 5E*), some of which at are ACHR+ cells (*Figure 5A″*). Similarly, sacral-derived cells that expressed TH+ (*Figure 5C and C″*, *Figure 5—figure supplement 2C–C″″*) and DBH+ (*Figure 5D and D″*, *Figure 5—figure supplement 1D–D″″*) cells mostly resided in the Nerve of Remak while the vagal-derived cells were located in the myenteric or submucosal plexuses (*Figure 5—figure supplement 1C–C″″*, TH; *Figure 5—figure supplement 1D–D″″*, DBH). Both vagal and sacral neural crest contributed to Schwann cells and progenitors/glia expressing SOX10 (*Figure 5E–E″*, *Figure 5—figure supplement 1E–E″″*, vagal; *Figure 5—figure supplement 2E–E″″*, sacral) and P0 (*Figure 5F–F″*, *Figure 5—figure supplement 1F–F″″*, vagal; *Figure 5—figure supplement 2F–F″″*, sacral). These results confirm that the sacral neural crest contributes to both a large portion of the Nerve of Remak as well as a subset of neurons in the post-umbilical gut at E10.

## Subclassification of vagal and sacral neural crest-derived neuronal cell types

To clarify contributions to neuronal subtypes, we extracted all cells from neuronal (*ELAVL4+*) clusters C2 and C4 (*Figure 4A′*) and re-clustered them into 11 subclusters (sC0-10) to identify genes characteristic of each cluster (Table S3; *Figure 6A*). We plotted expression of receptors, neurotransmitters, and neuropeptides (*Figure 6B and C*) and generated a table of transcription factors associated with each cluster (Table S4). sC0, sC1, and sC9 were identified as neuroblast-like due to high expression of the neural crest genes *SOX10* (*Kim et al., 2003*) and *ZEB2*, plus *ASCL1* (*Castro et al., 2011*), *PENK*, *NEFM*, *GAL*, *CHGA*, *DDC*, and *CHRNA7*. sC5 has similar gene expression to sC0 and sC1 but was considered to be immature neurons undergoing neurogenesis due expression of *Ascl1* and transcription factors *ETV5* (*Liu and Zhang, 2019*) and *ETV1* (*Wright et al., 2021*).

sC2 represents neurons that are *CHRNA7/GAL/DLX6* (*Vohra et al., 2006*)/*HMX3* (*Heanue and Pachnis, 2006*) positive; however, the expression of *ASCL1* points to an immature population. sC10 expresses *CHAT*, *TAC1*, *PENK*, *CALB2*, and *MEIS1,* associated with neurogenesis and neural crest invasion in zebrafish ENS development (*Uribe and Bronner, 2015*). sC3 is similar to sC10, with expression of *CHAT/TAC1/CHRNA7/MEIS1*, but also expresses *GAL, VIP, NPY, MYTL, ZFHX3*. Only low expression of the nitrergic gene *NOS1* was observed. Interestingly, both sC10 and sC3 express *PBX3*, a transcription factor that has been previously linked to postmitotic interneruons in mice (*Morarach et al., 2021*), but whose role in chick remains unknown. sC6 and sC7 express *CHAT*, *NEFM*, *CALB1*, *ISL1*. sC7 is marked by expression of *SST, DDC, NEFM*, and *GAL,* whereas sC4 is similar to sC7 except for low expression of *SST* and absence of catecholaminergic genes (*DBH/TH*). sC8 expresses the catecholaminergic genes *DBH* and *TH*, and the serotonergic genes *SLC18A2* and *DDC*.

We next separated each population into separate UMAPs of vagal pre-umbilical gut, vagal post-umbilical gut, and sacral post-umbilical gut (*Figure 6D*). Additionally, we calculated the proportion of cells in each cluster contributed by each population (*Figure 6—figure supplement 1*). As predicted by each population's UMAP of all RIA+ cells (*Figure 2C*), there is a distinct distribution pattern across the subclusters based on population of origin and gut location. There are no subclusters that are exclusively sacral-derived, while sC10 (*CALB2/TAC1/PBX3*) is comprised of only vagal cells (50% post- and pre- umbilical).

Comparing pre-umbilical and post-umbilical populations, we find no clusters that are unique to the pre-umbilical environment. However, the pre-umbilical vagal does heavily contribute to the putative

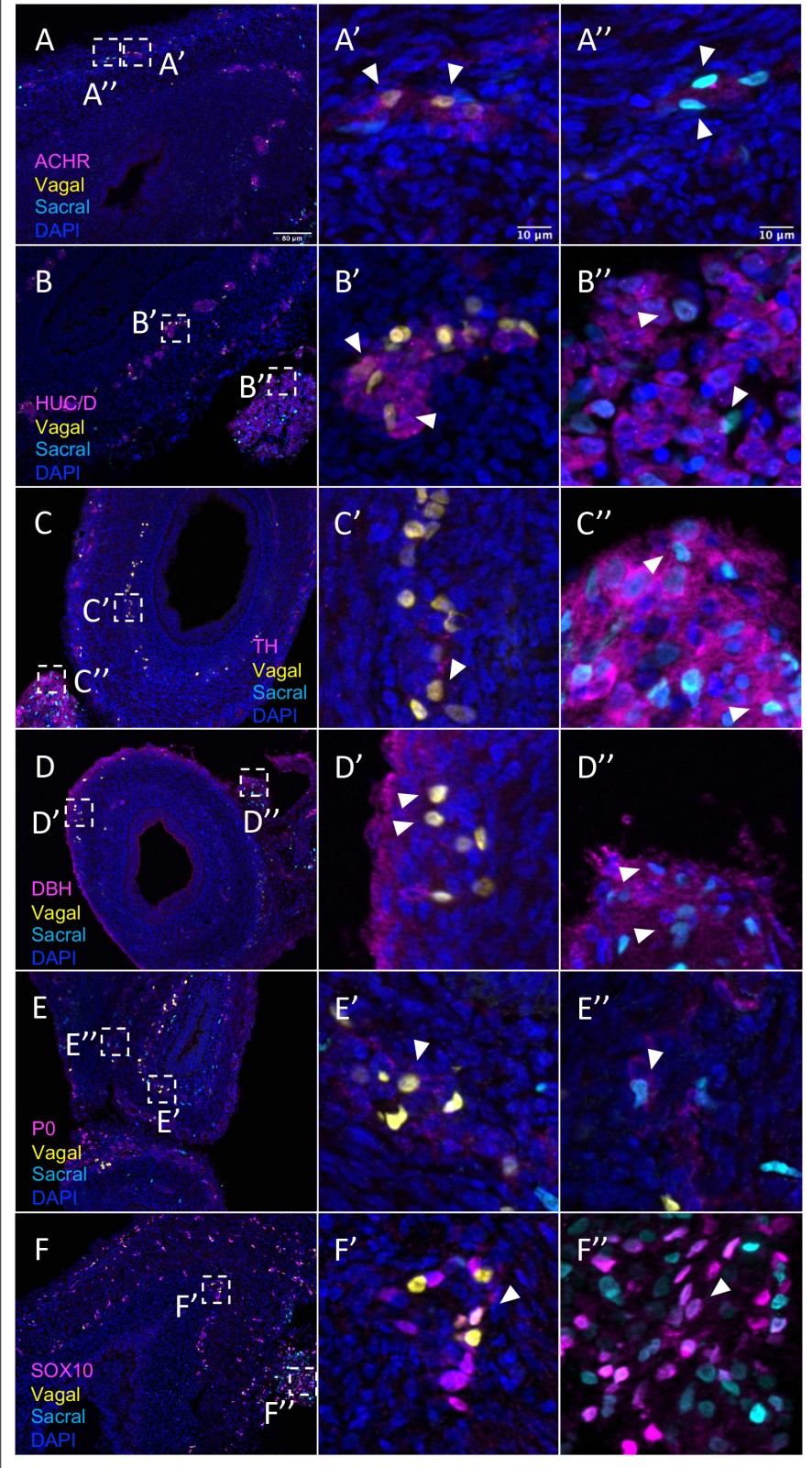

**Figure 5.** Relative contributions of vagal and sacral neural crest cells to the post-umbilical gut. (**A**) Acetylcholine receptor was broadly expressed by vagal neural crest along the post-umbilical gut (inset **A'**); ACHR was also present in sacral neural crest-derived cells (inset **A"**). (**B, C**) Differentiated neuronal markers HUC/D and TH were expressed by vagal (insets **B'**, **C'**) neural crest cells in the hindgut, while sacral neural crest cells were most

*Figure 5 continued on next page*

*Figure 5 continued*

present in the Nerve of Remak (insets **B″**, **C″**). (**D**) DBH expressing vagally derived cells were observed primarily in the myenteric plexus of the hindgut while sacral-derived DBH+ cells were predominantly in the Nerve of Remak (**D″**). (**E**) Both populations contributed P0+ glial cells within the plexuses of the hindgut (insets **E′–E″**). (**F**) SOX10+ progenitors were derived from both populations with vagal cells residing within the hindgut (inset **F′**) and sacral cells predominantly located within the Nerve of Remak (inset **F″**). Insets (**A′–F′, A″–F″**) show magnified regions in the corresponding dashed box. White arrows indicate double-positive cells. Scale bars for main figure (**A–F**): 80 µm. Scale bars for insets: 10 µm.

The online version of this article includes the following figure supplement(s) for figure 5:

**Figure supplement 1.** Relative contributions of vagal neural crest cells to the post-umbilical gut separated by channel.

**Figure supplement 2.** Relative contributions of sacral neural crest cells to the post-umbilical gut separated by channel.

progenitor subclusters 0 and 5 (88 and 83%, respectively) compared to the post-umbilical vagal (9%, 15%). This pattern is also seen in sC3 (*GAL/VIP/NPY+*) and sC2 (*CHRNA7/GAL+*), in which the post-umbilical gut cells form only 23 and 15%, respectively. There is minimal sacral contribution to these clusters. Conversely, sC4 (*CHRN7A/NEFM+*), sC6 (*CALB1/CHAT+*), and sC7(*SST/DDC+*) are unique to the post-umbilical environment and interestingly are comprised of over 50% sacral neural crest. Although not exclusive to the post-umbilical environment, *SOX10+* putative neuronal progenitor sC1/sC9 and sC8 (*TH/SLC18A2/DDC/SST+*) have higher percentages of sacral cells (72%, 83%, 66%) and limited contribution from the post-umbilical vagal (27%, 10%, 33%).

## RNA velocity analysis reveals developmental trajectories leading to ENS differentiation

In order to better identify potential differences in vagal and sacral contributions to neuronal and glial lineages within the E10 pre- and post-umbilical gut, we isolated cells from neuronal (C2, C4), glial (C1, C6, C8), and progenitor clusters (C0, C7, C3) (defined by clustering of RIA+ cells presented in *Figure 2*) for RNA velocity analysis (*La Manno et al., 2018*). This method utilizes the ratio of spliced and unspliced mRNA to infer information regarding terminal cell state and the cell's experience of latent time. Using a dynamical model in *scVelo* (*Bergen et al., 2020*), we determined the predicted trajectory for each subpopulation (*Figure 7A–C*). Both pre- and post-umbilical vagal-derived populations have C0 as a progenitor pool with developmental trajectories reaching into neuronal and enteric glial clusters (*Figure 7A and B*). Interestingly, sacral-derived cells lack this progenitor cluster at E10 (*Figure 7C*). Instead, the sacral neural crest cells have differentiation trajectories within enteric glial/SCP or neuronal clusters. The enteric glial clusters within the vagal post-umbilical also have intertwined trajectories including into the putative stem cell cluster C3 indicating potential plasticity between the clusters (*Figure 7B*). This finding is supported by a recent study that demonstrates the ability of enteric glia to reenter a progenitor-like state and undergo neurogenesis in vitro and in an injured gut (*Laddach et al., 2023*). We also noted that in both the vagal- and sacral-derived post-umbilical cell populations, there are RNA velocity trajectories from neuronal clusters C2 into C4 which may reflect previously reported post-mitotic neuronal differentiation (*Morarach et al., 2021*).

Further analysis was performed using *CellRank* (*Lange et al., 2022*), a method that builds upon the *scVelo* analysis to reconstruct single-cell dynamics in populations without a known developmental trajectory and predicts terminal states. We calculated the probability for each cell to give rise to each respective terminal state ('absorption probability' of the Markov Model) (*Figure 7D*). Based on this analysis, we observed that the vagal-derived cells in the pre-umbilical gut have a terminal state in the neuronal cluster C2 with high probability of contribution from a subsection of the progenitor pool (C0). C0 was also identified as its own terminal state in the pre-umbilical vagal cells with the highest probability in a subsection of the cluster itself, pointing to self-renewal of the progenitor population. In contrast, vagal cells in the post-umbilical state have the neuronal cluster C4 as terminal state with high probability of contribution from progenitor clusters (C0, C3, C7). Interestingly, this arises from neuronal cluster C2. Like the pre-umbilical vagal cells, a self-contributing progenitor pool (C3) was also identified as a terminal state. Sacral-derived cells have terminal states of neuronal cluster C4 and enteric glial cluster (C8). The terminal fate C4 has the highest contribution from itself while the

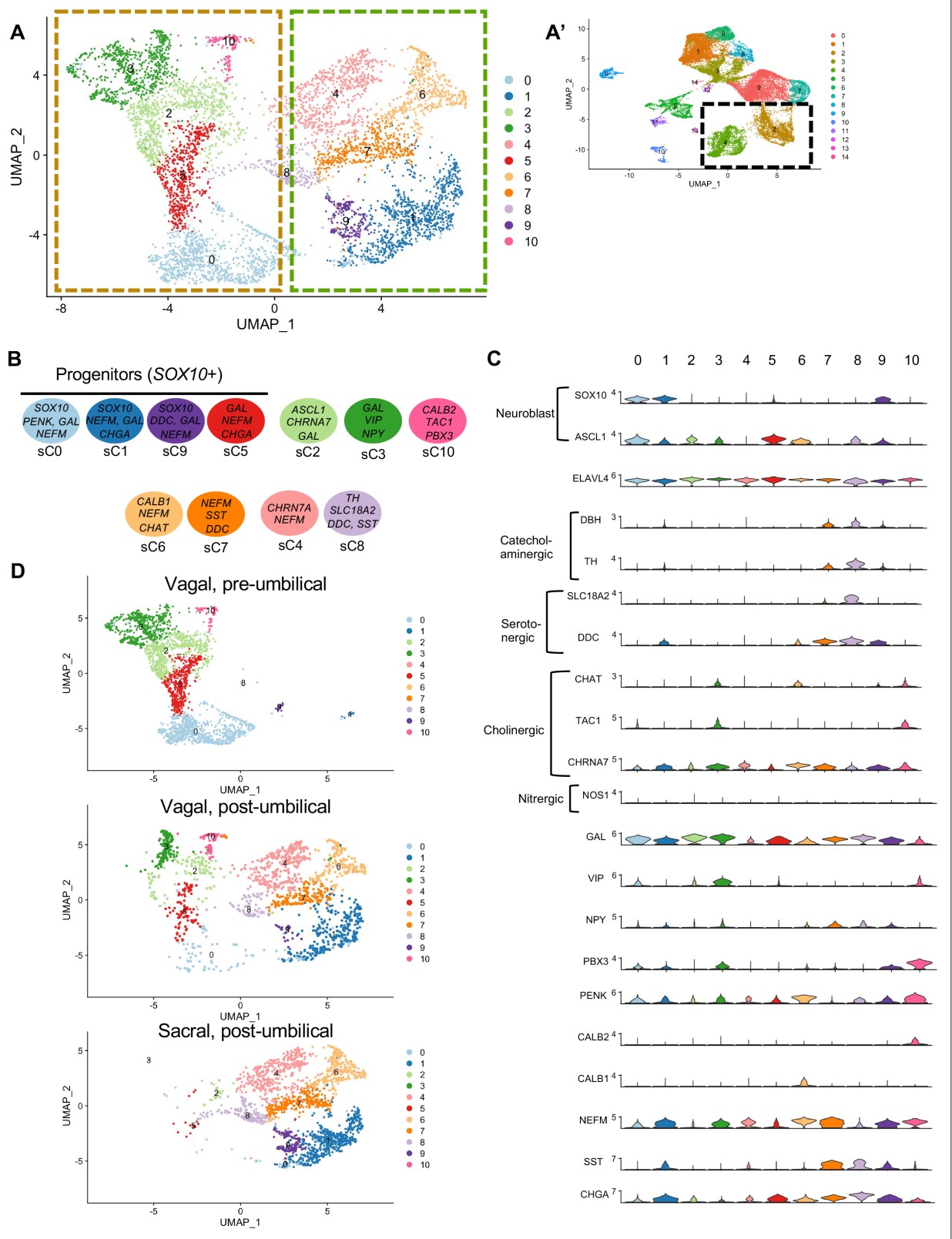

**Figure 6.** Subclassification of neuronal clusters. (**A**) Subclustering of neuronal clusters (resolution of 0.4) (c4, c2 inset **A'**) resulted in 11 distinct populations (sc0-10). (**B**) Schematic diagram demonstrating marker genes of each cluster. (**C**) Violin plots of neurotransmitters, neuropeptides, receptors, and key genes representing neuroblasts and specific neuronal functions such as catecholaminergic, serotonergic, cholinergic, and nitrergic. (**D**) UMAP representation (resolution 0.4) of each population's subclustered neuronal cells.

*Figure 6 continued on next page*

*Figure 6 continued*

The online version of this article includes the following figure supplement(s) for figure 6:

**Figure supplement 1.** Proportion of pre-umbilical vagal, post-umbilical vagal, and sacral populations across neuronal subclusters.

enteric glial terminal state has high probability of contribution from other enteric glial clusters (C1, 6) and a putative stem cell cluster (C3). Taken together, this analysis reveals potential differences in the precursor pool of sacral versus vagal neural crest derived cells, with the sacral neural crest giving rise to 'enteric glia' (*Laddach et al., 2023*) that potentially reflect a transitional state that retains both glial and neuronal potential. This highlights differences in developmental potential and timing of differentiation between these two populations.

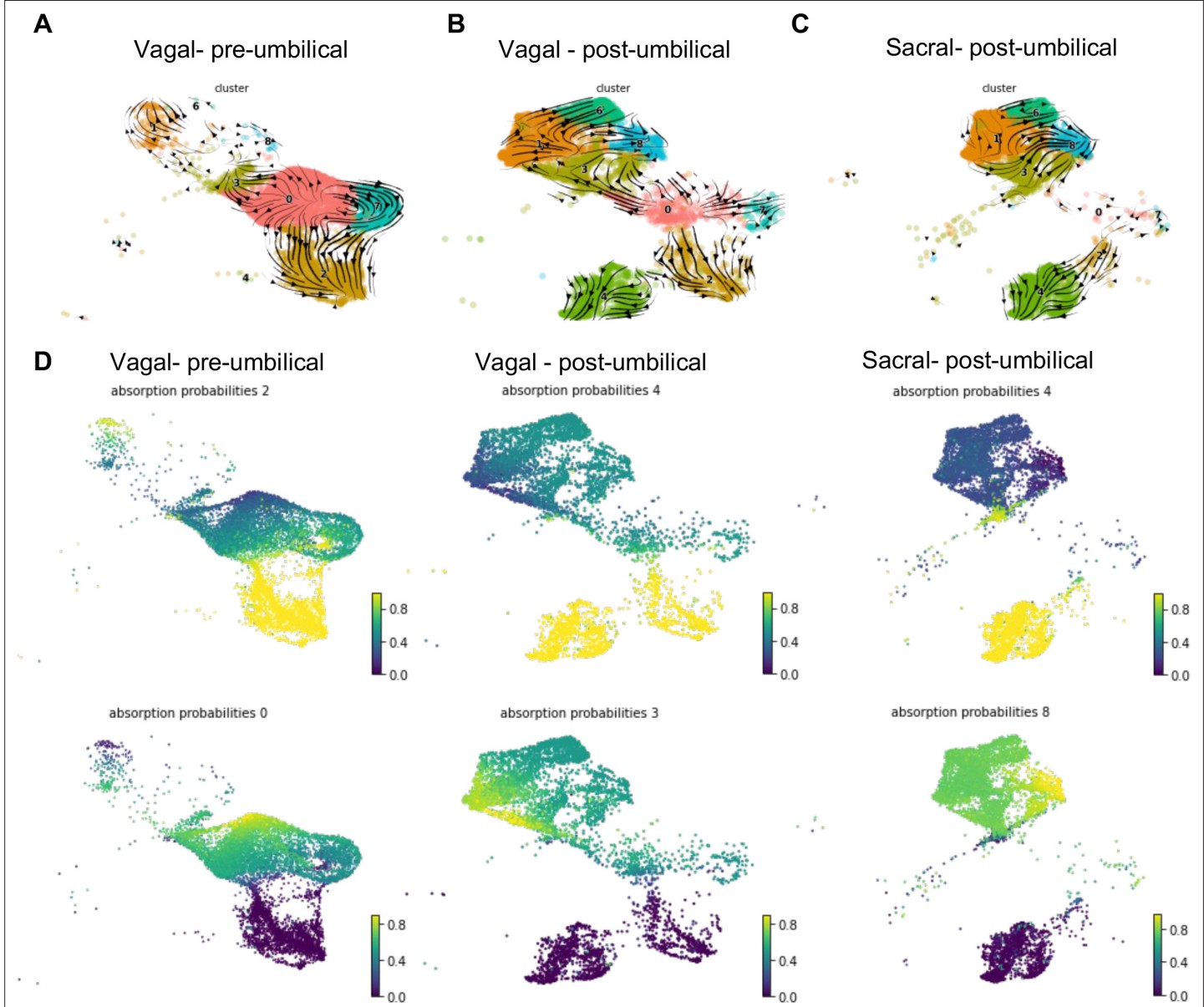

**Figure 7.** RNA velocity and terminal fate analysis of vagal and sacral neural crest cells in the enteric nervous system (ENS). (**A**) Streamlines of RNA velocity projected on UMAP for pre-umbilical vagal-derived cells. (**B**) Streamlines of RNA velocity projected on UMAP for post-umbilical vagal-derived cells. (**C**) Streamlines of RNA velocity projected on UMAP for post-umbilical sacral-derived cells. (**D**) Absorption probabilities for the terminal fates determined in each population: pre-umbilical vagal-derived, post-umbilical vagal-derived, and post-umbilical sacral-derived.

## Validation of sacral contribution to enteric glial/Schwann cell precursor-like fates

Given that, in addition to neurons, the sacral neural crest has predicted terminal fates of enteric glial, we performed additional validation of glial/SCP-like contributions sections containing H2B-YFP (cyan in figure) labeled sacral neural crest. The results show that the sacral neural crest gives rise to enteric glial fates with numerous cells residing in the Nerve of Remak, proximal to the hindgut. Immunostaining detected co-expression of P0 and YFP (cyan) cells, indicating sacral neural crest contribution to an SCP population (*Jessen and Mirsky, 2005*; *Figure 8A and A'*, *Figure 8—figure supplement 1A–A""*), as well as PLP1 (*Figure 8B and B'*, *Figure 8—figure supplement 1B–B""*) a marker of glial cells that has been shown to be widely expressed by enteric glia in mice (*Rao et al., 2015*). Indeed, conditional removal of *Plp1* expressing enteric glia disrupts gastrointestinal motility in female mice (*Rao et al., 2017*), indicating the importance of such cells in ENS health. Additionally, we observed a small population of sacral neural crest cells expressing the canonical glial marker GFAP (*Jessen and Mirsky, 1980*; *Figure 8C and C'*, *Figure 8—figure supplement 1C–C""*). Together these results confirm our predicted fate at E10 of the sacral neural crest cells to enteric glia/SCP-like cells.

## Discussion

The enteric nervous system regulates critical gastrointestinal functions including digestion, fluid secretion, and immune interactions. Abnormal ENS development can lead to enteric neuropathies including Hirschsprung's disease, characterized by lack of motility and obstruction. Studies of ENS development have primarily focused on the role of the vagal neural crest in the ontogeny of ENS disorders and did not parse the respective contributions of sacral versus vagal neural crest (*Amiel et al., 2008*; *Kenny et al., 2010*). Thus, the role of the sacral neural crest, which colonizes the hindgut in close coordination with the vagal neural crest, has been largely understudied. To better understand the derivatives of the sacral neural crest and their coordination with the vagal neural crest during ENS development, here we examine the diversity of cell types arising from vagal versus sacral axial levels at single-cell resolution in the E10 embryonic chick gut.

Ablation and heterotopic grafting experiments previously have been used to study the interplay between the vagal and sacral populations but have led to contradictory interpretations. While some studies concluded that vagal and sacral neural crest exhibit autonomous migration properties independent of the environment, others suggested a role for environmental influences. In an aganglionic hindgut model created by surgically removing the caudal part of vagal neural crest, transplanted quail sacral neural crest cells migrated into the hindgut and produced a small number of neurons but failed to compensate for the absence of vagal-derived neurons. This suggested that sacral neural crest cells do not require the vagal population to migrate but lack intrinsic ability to populate the gut fully (*Burns et al., 2000*). Reciprocally, when vagal neural crest cells are grafted to the sacral region, they migrate earlier and produce a larger neuronal population than the endogenous sacral neural crest cells (*Burns and Le Douarin, 2001*). However, other studies suggested a more prominent environmental effect, such that interchanged vagal and sacral neural crest cells migrated according to the local environment (*Erickson and Goins, 2000*). Consistent with this, combining chick gut before neural crest colonization with chick or quail neural crest revealed that sacral neural crest cells can colonize the colorectum independent of the vagal neural crest, but require the hindgut environment to differentiate (*Hearn and Newgreen, 2000*). Additionally, a recent study has demonstrated that human pluripotent stem cell (hPSC)-derived sacral neural crest are required together with hPSC-derived vagal neural crest to repopulate an aganglionic murine colon (*Fan et al., 2023*).

Our study using axial-level-specific labeling and transcriptomic analysis helps to resolve some of these apparent discrepancies. By utilizing RIA retroviruses within the chick embryo, we provide a complementary approach to address questions of developmental potential of vagal versus sacral neural crest population. RIA enables selective labeling of each population, facilitating comparison of the relative contributions of the vagal and sacral neural crest in the pre- and post-umbilical gut as well as the role of environmental factors therein. Coupling this neural crest axial-level-specific lineage labeling technique with transcriptomic analysis further provides granular detail of cell types within the developing ENS and differences in each neural crest population's derivatives.

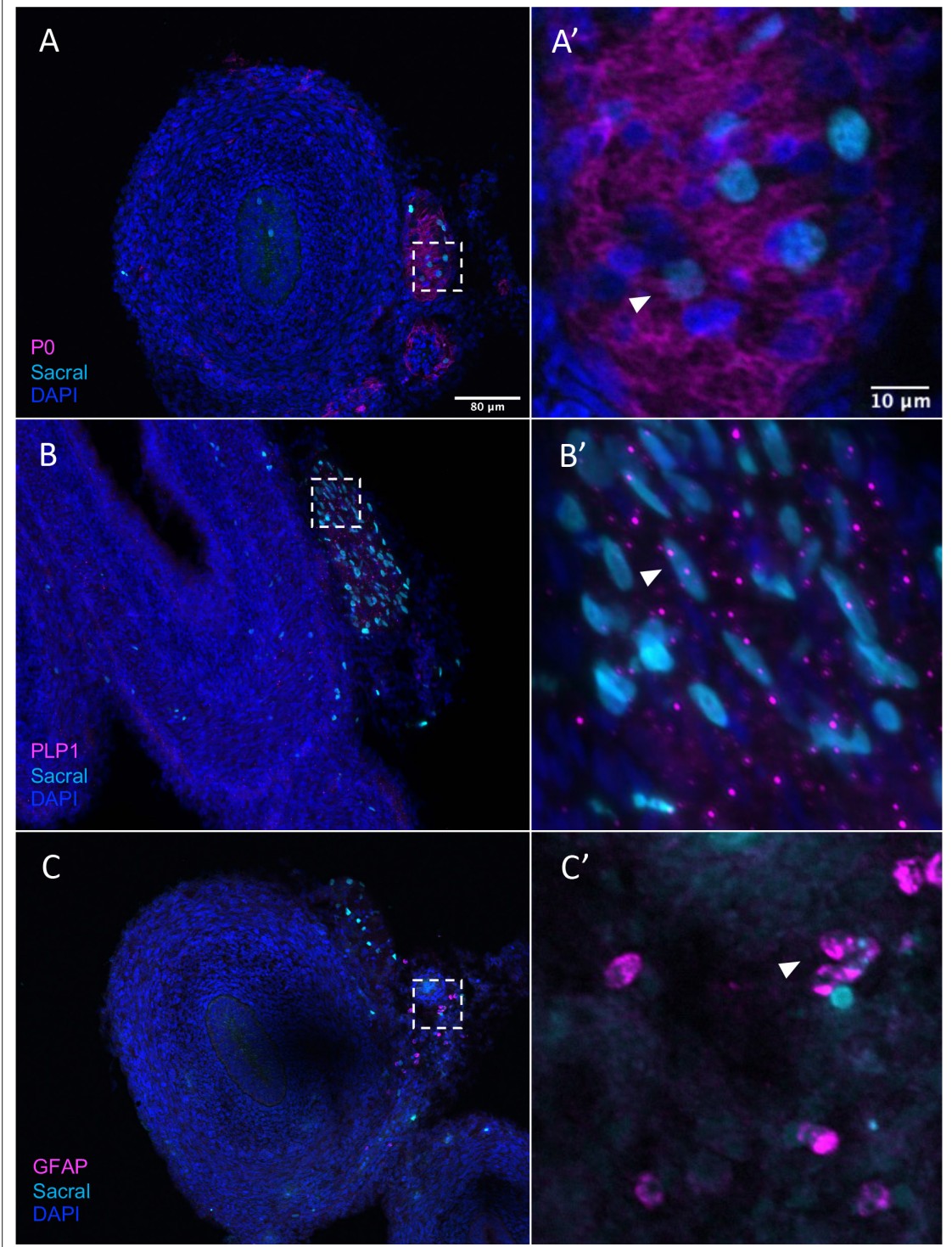

**Figure 8.** Sacral-derived glial fate in the Nerve of Remak. Sacral-derived cells contributed enteric glia within the Nerve of Remak labeled with P0 (**A, A'**), PLP1 (**B, B'**), and GFAP (**C, C'**). Insets (**A'–E'**) show magnified regions in the corresponding dashed box. White arrows indicate double-positive cells. Scale bars for main figure (**A–E**): 80 µm. Scale bars for insets (**A'–E'**): 20 µm.

The online version of this article includes the following figure supplement(s) for figure 8:

**Figure supplement 1.** Sacral-derived glial fate in the Nerve of Remak separated by channel.

Using bulk RNA-sequencing, we find some differences between sacral and vagal-derived ENS cell types at the population level at E10. For example, we find *TNC* expression to be specific to the vagal neural crest (*Figure 1C and D*), consistent with its requirement for migration into the hindgut by changing the extracellular microenvironment (*Akbareian et al., 2013*). In addition, the sacral neural crest expresses high levels of SOX10-mediated *Cdh19* (*Huang et al., 2022*), as well as *Pax3* and 5-HT3 for innervation and neuronal maturation in the pelvic ganglion (*Deal et al., 2021*; *Ritter et al., 2021*). *Ret* is also known to be upregulated in vagal neural crest cells to mediate more invasive behavior than in sacral neural crest (*Delalande et al., 2008*).

Our single-cell RNA-sequencing results demonstrate that the pre-umbilical gut is solely populated by the vagal neural crest, giving rise predominantly to motor neurons and fibroblasts. In the post-umbilical gut, vagal neural crest cells form enteric glia, several neuronal subtypes, and some non-neural fates. Interestingly, the sacral and vagal neural crest contributions to the post-umbilical gut overlap, indicating the importance of the environment in post-umbilical ENS development. Comparison of the two gut axial levels demonstrates that only the post-umbilical gut has the unique neuronal subclusters sC4 (*CHRN7A/NEFM+*), sC7 (*SST/DDC+*), and sC6 (*CALB1/CHAT+*). Interestingly, all of these unique post-umbilical clusters have large contribution from the sacral neural crest (>50%) (*Figure 6—figure supplement 1*).

Importantly, our data reveal large differences in cell-type contribution between vagal neural crest-derived cells that localize in the pre-umbilical versus post-umbilical gut. Indeed, the complement of post-umbilical vagal-derived clusters is more similar to the sacral's than the profile of pre-umbilical vagal-derived clusters. Within the post-umbilical gut, there is significant overlap between vagal and sacral neural crest fates. This suggests that there may be a relatively uniform developmental potential between vagal and sacral neural crest cells and that the local environment of the hindgut may play an important role in guiding their differentiation.

Indeed, the data suggest that there is no population in the E10 chick post-umbilical ENS that is exclusively derived from the sacral neural crest; however, the sacral does contribute over 50% of cells to the glial clusters C1/8 and non-neural clusters C11/12/13 (*Supplementary file 3*). We identify multiple clusters derived predominantly from the vagal populations, including two progenitor clusters (C0/C7) and the *GAL/VIP/NPY+* (sC3) and *CALB2/TAC1/PBX3+* (sC10) neuronal subclusters. The absence of sacral contribution to these neuronal subtypes may be the result of differences in timing of differentiation, but may also partially explain the inability of the sacral neural crest to compensate for loss of the vagal (*Burns et al., 2000*).

RNA velocity analysis demonstrates that the vagal neural crest maintains a glial/neuronal progenitor pool while the sacral neural crest does not at E10. While the presence of a sacral-derived neuron cluster and the absence of a putative neuroblast pool could be interpreted as early termination of sacral neurogenesis, we posit there may be further sacral contributions to ENS neurons at later time points from enteric glia/SCPs that may later migrate into the gut wall from the Nerve of Remak. This is consistent with a recent study that shows the potential for enteric glia to regain progenitor-like state and undergo neurogenesis in vitro and in an injury model (*Laddach et al., 2023*).

Previous studies have suggested that the maintenance of proliferative capacity in the 'wavefront' of invading vagal neural crest is critical for successful colonization of the elongating gut (*Nagy and Goldstein, 2006*; *Landman et al., 2007*; *Simpson et al., 2007*). Consistent with this possibility, endothelin-3, a gene implicated in Hirschsprung's disease (*Bidaud et al., 1997*; *Kenny et al., 2000*), is important for maintaining a pro-proliferative environment for the vagal neural crest in the avian hindgut (*Nagy and Goldstein, 2006*) and is required for sacral neural crest colonization in the murine hindgut (*Baynash et al., 1994*).

RNA velocity further suggests distinct developmental trajectories for each neural crest population. Vagal-derived cells have a terminal state of *VIP/TAC1+* neurons in the pre-umbilical gut and *DDC/PNMT+* neurons in the post-umbilical gut similar to the predicted terminal neuronal state of sacral-derived cells (*Figure 7C*). This highlights the possible importance of the local environment in ENS development. The high probability of contributions of *VIP/TAC1+* neurons (C2) and *DDC/PNMT+* neurons (C4) (*Figure 7C*) is similar to the murine data demonstrating post-mitotic differentiation mediated by the transcription factor *Pbx3* (*Morarach et al., 2021*). We also see high expression of *PBX3* in the putative motor neuron subcluster sC10 (*Figure 6C*), thus indicating a homologous role for this transcription factor in chick ENS development.

In addition to *DDC*/PNMT+ neurons, the sacral neural crest is predicted to give rise to enteric glia or SCPs (*Figure 7C*), confirmed by our immunohistochemical analysis (*Figure 8*). We noted a large number of sacral-derived enteric glial cells residing in the Nerve of Remak, a structure unique to avian embryos (*Goldstein and Nagy, 2008*). Original studies by Burns and Le Douarin found that the sacral neural crest cells form the Nerve of Remak at E3 and continue to reside there until entering the gut between E8 and E10 (*Burns and Douarin, 1998*), possibly due to the presence of the chemorepellent SEMA3A in the hindgut (*Shepherd and Raper, 1999*). Quail-chick chimera indicated that Nerve of Remak alone may not be sufficient to populate the hindgut; in mice, the pelvic plexus represents the staging area for sacral neural crest cells to populate the gut (*Nagy et al., 2007*).

Our single-cell RNA-sequencing identified an enteric glial cluster (C8) that also expresses the canonical SCP marker *MPZ* (P0). Both vagal and sacral neural crest contribute to this cluster in the post-umbilical gut, as confirmed by the expression of P0+ (*Figures 7C and 8A*). However, there is a paucity of putative SCPs in the pre-umbilical gut. Studies in chicken and mice have shown that the vagal neural crest consists of a subpopulation of cells that take on a SCP fate prior to enter the gut, migrating along extrinsic nerves, contributing a stem cell pool that undergoes neurogenesis. Vagal-derived SCPs (emerging from the neural tube adjacent to somites 1–2) in chick migrate into the pre-umbilical gut via the vagus nerve and innervate the esophagus (*Espinosa-Medina et al., 2017*). In mice, sacral-derived SCPs have been shown to migrate into the hindgut via the pelvic nerve, forming ~20% of neurons in the colon (*Uesaka et al., 2015*). Additionally, a subset of SCPs derived from somite level 3–7 migrate ventrally and invade the esophageal mesenchyme (*Espinosa-Medina et al., 2017*). As the labeling technique used in this study would be expected to label both early migrating neural crest cells as well as neural crest-derived SCPs, we cannot distinguish whether the putative SCPs observed at E10 are derived from a neural crest population that initially migrates into and invades the gut or a later migrating SCP population.

Previous studies in mouse, human, and zebrafish, have provided valuable information regarding gene expression, cell fate divergence, and neuronal subtypes within the ENS (*Morarach et al., 2021*; *Lasrado et al., 2017*; *Memic et al., 2018*; *Drokhlyansky et al., 2020*). By focusing transcriptome analysis on the myenteric plexus of the small intestine at postnatal day 21, a study in mice identified 12 distinct neuronal classes categorized by a combination of neurotransmitters. Our results have identified most neurotransmitter genes found in the murine system with the exception of NOS1+ nitrergic neurons, GAD2+ GABAergic neurons, or SLC17A6+ Glutamatergic neurons (*Morarach et al., 2021*), which may develop at a later time point than studied here (*Figure 7B*). Another study, utilizing RAISIN RNA-seq, identified 21 neurons and 3 glial clusters in the mouse small intestine and colon and 14 neuronal clusters in the human colon. The neuronal subsets we have identified generally agree with this study, showing an overlap with murine *Tac1*/*Chat*+ neurons (sC10) and *Penk*+ neurons (sC6) (*Figure 6C*; *Drokhlyansky et al., 2020*). Rather than a *Nos1*+ inhibitory motor neurons, we identified a *Gal*/Vip +expressing neuronal cluster (sC3). We do observe similarities between the murine and chick glial clusters, both of which have *Pmp22/Frzb/Cdh19/Plp1*+ glial clusters (C1/6/8) (*Drokhlyansky et al., 2020*). Similarly to studies done in zebrafish embryos, which observed enteric neural crest progenitors labeled with *sox10*, *phox2bb*, and *zeb2a*. Similar to our study, *vip*+ and *pbx3*+ neuronal subtypes were identified in later zebrafish stages alongside neural crest-derived melanocytes and mesenchyme in the posterior section of the larvae (*Howard et al., 2021*).

While previous studies did not separate vagal from sacral contributions, the small intestine and colon contain both sacral and vagal populations, which are distinct from anterior vagal derivatives (*Figure 2C*). Our analysis reveals that gene expression patterns are markedly different between the vagal-derived cells in the pre- and post-umbilical gut (*Figure 2C*), confirming the results from previous studies within proximal versus distal colon (*Drokhlyansky et al., 2020*). This important conclusion suggests that the ENS is not uniformly distributed throughout the gut but varies from proximal to distal. Work in chick has identified a gene regulatory network (GRN) of Tfap/Sox/Hbox/bHLH transcription factors that determines vagal neural crest fate (neural, mesenchymal, or neuronal) prior to delamination and subsequent contribution to the ENS in the pre-umbilical region (*Ling and Sauka-Spengler, 2019*). Whether the same GRN regulates the vagal or sacral neural crest contribution to the post-umbilical gut remains to be determined.

Taken together, the present results suggest that there are different developmental programs for vagal versus sacral neural crest population. Our results may help explain why the sacral neural crest

cannot completely compensate for the loss of vagal neural crest. The cell composition of the post-umbilical ENS is distinct from that of pre-umbilical ENS, with major differences resulting from the differential contributions of the sacral neural crest. In addition, the differentiation program of vagal-derived neural crest in the post-umbilical gut is different from that of the pre-umbilical vagal population, suggesting that environmental factors have a large influence on cell fate.

## Methods

### Retroviral labeling and chick embryology

H2B-YFP (#96893) and H2B-RFP (#92398) obtained from Addgene were cloned into the RIA plasmid between Not1 and Cla1 sites. RIA-H2B-YFP/RFP was transfected into DF1 cells (ATCC, Manassas, VA; #CRL-12203, Lot number 62712171, Certificate of Analysis with negative mycoplasma testing available at ATCC website) using PEI standard transfection protocol. DF1 cells were maintained in Gibco Dulbecco's Modified Eagle Medium (DMEM) supplied with 10% FBS for 4 d, with 12 ml of supernatant collected per day. The supernatant was concentrated using ultracentrifuge for at $76,000 \times g$ for 1.5 hr to get a viral stock tittered about $10^7$ pfu/ml, aliquoted, and stored at $-80°C$ until use. Viral solution was supplemented with 0.3 μl of 2% food dye (Spectral Colors, Food Blue 002, C.A.S# 3844-45-9) as an indicator, injected to fill the neural tube between somite 1–7 at HH10 to label vagal neural crest and/or posterior to somite 28 at HH17 to label sacral neural crest in ovo. Embryos were supplied with Ringer's Solution (0.9% NaCl, 0.042% KCl, 0.016% $CaCl_2 \bullet 2H_2O$ wt/vol, pH 7.0), sealed with surgical tape, and incubated at 37°C until embryonic day 10.

### Vital dye labeling of the sacral neural crest, sectioning, and imaging

Vital dye DiI (Invitrogen #V22885) was diluted 1:5 with 10% sucrose and injected into HH17 embryos in ovo posterior to somite 28. Embryos were supplied with Ringer's Solution (0.9% NaCl, 0.042% KCl, 0.016% $CaCl_2 \bullet 2H_2O$ wt/vol, pH 7.0), sealed with surgical tape, and incubated at 37°C until subsequent collection. Timepoint (T) 0 embryos were collected after 1 hr of incubation. T24 embryos were collected 24 hr after injection. Embryos were fixed in 4% PFA in PBS (pH 7.5) for 30 min at room temperature and washed with PBS for three times. Fixed samples were incubated in a gradient of sucrose (5% for 2 hr at room temperature, 15% at 4°C overnight) and in gelatin at 37°C overnight. Embryos were embedded in gelatin solution, flash-frozen with liquid nitrogen, and mounted with Tissue-Tek O.C.T compound (Sakura #4583) for sectioning (Microm HM550 cryostat). Embryo sections were incubated in 1× PBS at 42°C until the gelatin was dissolved and stained with DAPI (CAT #) before being mounted with Fluoromount-G Mounting Medium (Invitrogen # 00-4958-02).

All sections were imaged with Zeiss AxioImager.M2 with Apotome.2. Images were cropped and magnified for representation.

### Gut cell dissociation and fluorescence-activated cell sorting (FACS)

At embryonic day 10, the gastrointestinal tract, including the associated Nerve of Remak and pelvic plexuses (*Supplementary file 2*), was dissected from chick embryos and washed with Ringer's solution. Pre- and post-umbilical regions were separated, broke into pieces in chilled DPBS, and loose-fit homogenized in Accumax solution (EMD Millipore). 400 μl of Accumax-tissue mixture was aliquoted into 1.7 ml Eppendorf tubes and shaken at 37°C for 12 min. After dissociation, chilled Hanks Buffered Saline Solution (HBSS) supplemented by BSA (125 mg in 50 ml, Sigma; 0.2% w/v) and 1 M HEPES (500 μl in 50 ml, pH 7.5, Thermo Fisher) was added to quench the reaction. The dissociated cells were passed through a 70 μm cell strainer (Corning) and collected by centrifuging at $500 \times g$ for 11 min at 4°C. The cells were resuspended in HBSS-BSA, supplemented 7-AAD Viability Staining Solution (BioLegend # 420404, 500 TESTS), and sorted for YFP+, viable single cells using Sony SY3200 cell sorter at the Caltech Flow Cytometry Facility.

### Bulk transcriptome analysis

For vagal neural crest in the post-umbilical regions, sacral neural crest at post-umbilical regions, two biological replicates were processed, with each replicate containing YFP+ cells from three embryos. A total of 2000 cells per replicate were lysed to generate cDNA library using SMART-Seq v4 Ultra Low Input RNA Kit (Takara Bio). The library was sequenced with 50 million single-end reads with

50 bp length using HiSeq 2500 at the Millard and Muriel Jacobs Genetics and Genomics Laboratory Caltech. Sequencing reads were trimmed using *cutadapt* (**Martin, 2011**) and mapped to Galgal6 genome using *Bowtie2* (**Langmead and Salzberg, 2012**). *DESeq2* (**Love et al., 2014**) analysis was performed to find differential expressed genes between vagal and sacral neural crest at post-umbilical regions generated by *HTseq-count* (**Anders et al., 2015**). Differential gene expression was presented using Volcano Plot (coloring genes with Fold Change >2 and p-value <0.05) and Heatmap2 provided by the Galaxy platform. Because there were more upregulated genes in the sacral than vagal-post populations, we annotated genes related to neuronal function for sacral population, genes related to neuronal function, as well as genes with top fold-change and top significance in vagal-post population.

## Single-cell transcriptome analysis and data processing

For vagal neural crest in pre- and post-umbilical gut, sacral neural crest in post-umbilical gut, two biological replicates were obtained. Each replicate of pre-umbilical gut was pulled from three embryos; each replicate containing post-umbilical gut was pulled from six embryos. After FACS for viable YFP+ cells, 4600–5000 cells per replicate were used for library preparation by the SPEC at Caltech. The library was sequenced on NovaSeq S4 lane with 2 × 150 bp reads by Fulgent Therapeutics. To process fastq raw data, standard ENSEMBL galgal6 reference database was used. Single-cell level gene quantification was then performed using *Cellranger* v3.1.0 (**Zheng et al., 2017**) and *kallisto* 0.46.2 and *bustools* 0.40.0 pipelines (**Bray et al., 2016**) with default parameters. Gene count matrices from all the samples were combined and only cells with more than 200 genes detected were kept for the downstream analysis. To further remove potential doublet cells, *DoubletFinder* 2.0.3 package was used. Of these, cells with greater than 2 transcripts of the RIA retrovirus promotor mRNA were then selected for further analysis. Average number of RIA retrovirus promoter mRNA transcripts per cluster are presented in *Supplementary file 5*. This resulted in 26,993 cells total with an average of 10,261 transcripts/cell (median of 9701) and 2734 genes/cell (median of 2854). Gene counts were normalized and scaled using Seurat v3.2.0 (**Butler et al., 2018**). From the pre-umbilical vagal samples, 10362 RIA+ cells were analyzed with an average of 10,395 transcripts/cell (median of 10,569 transcripts) and an average of 2794 genes/cell (median of 2945). From the post-umbilical vagal, 6454 RIA+ cells were recovered with an average of 10,307 transcripts/cell (median of 8779) and 2712 genes/cell (median of 2746). The sacral consisted of 10,177 RIA+ cells, with an average of 10,095 transcripts/cell (median of 8923) and 2688 genes/cell (median of 2760). The first 30 principal components from principal component analysis (PCA) were used to find neighbors with *Findneighbors* function before cell clustering with *FindClusters* function (resolution = 0.3). UMAP dimensionality reduction was performed using RunUMAP function with uwot-learn selected for the parameter umap.method.

Subclustering was performed on cells in C2 and C4. The first 30 principal components from the PCA were used to find neighbors with *Findneighbors* function before cell clustering with *FindClusters* function (resolution = 0.4). UMAP dimensionality reduction was performed using RunUMAP function with uwot-learn selected for the parameter umap.method.

## RNA velocity analysis

Loom files containing spliced/unspliced transcript expression matrices for all cells were generated using the velocyto.py pipeline (**La Manno et al., 2018**). Loom files were then refined to cell IDs that remained after Seurat filtering and cutoff of >2 RIA transcripts (RIA+ cells) for trajectory analysis and concatenated with corresponding Seurat UMAP coordinates, colors, and cluster identity. *scVelo* (**Bergen et al., 2020**) dynamical modeling was then performed on all RIA+ cells and each individual population using default settings. Estimated velocities were then used for terminal state analysis in *CellRank* (**Lange et al., 2022**). Terminal states, initial states, and absorption probabilities were calculated using default settings.

## Immunohistochemistry and imaging

Gastrointestinal tracts were dissected and fixed in 4% PFA in PBS (pH 7.5) for 25 min at 4°C and washed with PBS for three times. Pre- and post-umbilical regions were separately incubated in 15% sucrose at 4°C overnight and in gelatin at 37°C for 2 hr. Gut segments were embedded in gelatin solution, flash-frozen with liquid nitrogen, and mounted with Tissue-Tek O.C.T compound (Sakura #4583) for sectioning (Microm HM550 cryostat). Gut sections were incubated in 1× PBS at 42°C until

the gelatin was dissolved, soaked in 0.3% vol/vol Triton-X100 in 1× PBS for permeabilization. Blocking buffer was prepared in 1× PBS with 5% vol/vol normal donkey serum and 0.3% vol/vol Triton-X100. Sections were incubated with primary antibody at 4°C overnight. Sections were washed with 1× PBS for 10 min and three times. After the washes, sections were incubated with secondary antibody for 45 min at room temperature. List of primary antibodies used: 1:20 chicken anti-AchR ratIgG2a, mAB270 (DSHB Antibody Registry ID: AB_531809); 1:500 mouse anti-HuC/D IgG2b (Invitrogen, Cat# A21271); 1:20 chicken anti-mouse P0 IgG1, IE8 (DSHB Antibody Registry ID: AB_2078498); 1:500 rabbit anti-Sox10 (Millipore, Cat# AB5727); 1:500 rabbit anti-Tyrosine Hydroxylase (Millipore, Cat# AB152); 1:500 rabbit anti-DBH (ImmunoStar, Cat# 22806); and 1:250 rabbit anti-PLP1 E9V1N (Cell Signaling Technology #28702). List of secondary antibodies used: 1:1000 donkey anti-mouse IgG2b 647 (Invitrogen A31571), 1:1000 goat anti-mouse IgG1 647 (Invitrogen A21240), 1:1000 donkey anti-rat IgG 647 (Abcam ab150155), and 1:1000 goat anti-rabbit IgG 647 (Invitrogen A21245).

All sections were imaged with Zeiss AxioImager.M2 with Apotome.2. Images were cropped and magnified for representation.

## Acknowledgements

This work was supported by 1R01DK133480 to MEB and F31 HD111287 to JLL We thank Drs. Igor Antoshechkin and Vijaya Kumar and the Millard and Muriel Jacobs Genetics and Genomics Laboratory at California Institute of Technology for their guidance and support in bulk RNA-sequencing. We thank Jamie Tijerina and Rochelle Diamond from the Beckman Institute Flow Cytometry Facility for their help with the FACS. We thank Dr. Sisi Chen, Jeff Park, Prof. Matt Thomson, and SPEC at Caltech for their dedicated support in optimization and guidance in single-cell RNA-sequencing. We thank Dr. Fan Gao and Bioinformatics Resource Center in the Beckman Institute at Caltech for guiding us through single-cell transcriptomic analysis. We appreciate the help from Prof. Carlos Lois for kindly sharing equipment with us to perform RIA concentration. We thank Dr. Michael L Piacentino, Dr. Erica J Hutchins, and Prof. Angelike Stathopoulos for the helpful discussion on the manuscript.

## Additional information

### Competing interests

Marianne E Bronner: Senior editor, *eLife*. The other authors declare that no competing interests exist.

### Funding

| Funder | Grant reference number | Author |
| --- | --- | --- |
| National Institute of Diabetes and Digestive and Kidney Diseases | R01DK13348 | Marianne E Bronner |
| Eunice Kennedy Shriver National Institute of Child Health and Human Development | F31HD11128 | Jessica Jacobs-Li |

The funders had no role in study design, data collection and interpretation, or the decision to submit the work for publication.

### Author contributions

Jessica Jacobs-Li, Data curation, Formal analysis, Validation, Writing - review and editing; Weiyi Tang, Conceptualization, Data curation, Formal analysis, Validation, Investigation, Visualization, Methodology, Writing - original draft, Writing - review and editing; Can Li, Formal analysis, Methodology; Marianne E Bronner, Conceptualization, Resources, Funding acquisition, Writing - original draft, Project administration, Writing - review and editing

### Author ORCIDs

Jessica Jacobs-Li http://orcid.org/0000-0003-1339-3555

Weiyi Tang http://orcid.org/0000-0002-1279-1001

Marianne E Bronner http://orcid.org/0000-0003-4274-1862

**Decision letter and Author response**
Decision letter https://doi.org/10.7554/eLife.79156.sa1
Author response https://doi.org/10.7554/eLife.79156.sa2

## Additional files

### Supplementary files

• Supplementary file 1. Gene markers associated with single-cell RNA-seq clusters of vagal and sacral neural crest-derived cells.

• Supplementary file 2. Signaling pathways and transcription factors enriched in vagal and sacral neural crest-derived cells.

• Supplementary file 3. All cells from neuronal clusters C2 and C4 were re-clustered them into 11 subclusters (sC0-10).

• Supplementary file 4. Transcription factors associated with each cluster.

• Supplementary file 5. Average number of RIA retrovirus promoter mRNA transcripts per cluster.

• MDAR checklist

### Data availability

Sequencing data has been deposited to GEO, accession number GSE242228.

The following dataset was generated:

| Author(s) | Year | Dataset title | Dataset URL | Database and Identifier |
|---|---|---|---|---|
| Jacobs-Li J, Tang W, Li C, Bronner ME | 2023 | Single-cell profiling coupled with lineage analysis reveals vagal and sacral neural crest contributions to the developing enteric nervous system | http://www.ncbi.nlm.nih.gov/geo/query/acc.cgi?acc=GSE242228 | NCBI Gene Expression Omnibus, GSE242228 |

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
