## [Editor Report]

This paper is useful for researchers in the field of enteric neuroscience and peripheral nervous system development. The single cell RNA-sequencing based analysis of the developing chicken ENS, demonstrates differential cell identity contribution from the sacral and vagal neural crest and influence of the local distal embryonic environment for final differentiation. A basic classification scheme of neuronal cell types in the chicken combined with analysis of a more mature embryonic stages and functional data will however be needed in the future to determine the role of differential stem cell origin for final neuronal composition in the distal gut of chicken.

---

## [Decision Letter]

**Decision letter after peer review:**

Thank you for submitting your article "Single-cell profiling coupled with lineage analysis reveals distinct sacral neural crest contributions to the developing enteric nervous" for consideration by *eLife*. Your article has been reviewed by 2 peer reviewers, and the evaluation has been overseen by a Reviewing Editor and Kathryn Cheah as the Senior Editor. The following individual involved in review of your submission has agreed to reveal their identity: Ulrika Marklund (Reviewer #2).

Essential revisions:

1) Extend the strategy of the scRNA study to stages when ENS neurons from both vagal and sacral origin are fully differentiated, compared to the current data set in which the expression profiles indicate that the differentiation states of both populations do not match, including relatively immature cell states. This will allow solid conclusions about cell types contributed by either source of neural crest, as well statements of relative proportions of the different populations.

2) Validate the strategy for neural crest lineage tracing using the RIA retroviruses, given the substantial contribution of non-ENS cell types, such as hematopoietic cells, muscle, endothelial cells, etc.. This would be necessary to support the conclusion that the neural crest contributes to various gut cell types, which is potentially interesting given that multiple corresponding neural crest tracing studies in mouse label only limited or none of these cell populations.

Related to this point, it is important to show how far the injected virus spreads, given the viral labelling technique is new for avians.

3) Extending the bio-informatic analyses will be necessary to clarify cell type identities as well as the contribution of specific cell populations from the sacral and vagal neural crest, and thereby essential for supporting the conclusions of the study. Specifically, the following points should be addressed:

a) In addition of the merged datasets, provide separate analyses of the (pre- and postumbilical), and sacral (postumbilical) populations to elucidate the differentiation paths and relations of individual clusters with relevant methodologies (e.g. Velocity).

b) Comparison to existing ENS data sets from other species, in particular mammals, will likely help to define the various ENS populations in the chicken and place the results and conclusions of the study into the larger context of the ENS field. (recommended).

c) Related, this is the first scRNAseq study of the chicken ENS at this stage and will therefore represent a valuable resource to the ENS field across all species. Including for instance tables listing the DEG would be a highly useful supplement to the paper.

4) Validate the scRNAseq data and resulting conclusion by providing more extensive analyses of the different ENS populations employing immunohistochemistry similar to Figures 5 and 6, to i) define neuronal subpopulations and ii) quantify the size of the respective populations, which will support the comparison of the contributions from vagal and sacral neural neural crest and the conclusion that cells originating from different axial origin contribute different fates.

Central to this validation is to consider naming clusters systematically after prominent marker genes rather than possible function, since the physiology of the chicken ENS is currently not well characterised. This will allow for better comparison with other species.

[for detailed suggestions see comments by reviewer2]

5) Depending on the outcome of the above points, it will be important for the authors to choose an appropriate hypothesis to address functionally to corroborate the implication(s) of the promising scRNAseq results.

Additional points to be addressed:

i) Clarify whether there are true melanocytes in the GI-tract preparations or cell populations with a largely similar expression profile.

ii) Discuss the present results from chicken in the context of the work, with respect to neural crest-derived cell attaching to nerve bundles and obtaining a schwann cell precursor-like identity before migrating into the gut, from Uesaka et al. 2015 and Espinosa-Medina et al. 2017.

iii) use distinguishable colours when representing more than one antibody staining in data panels showing immunohistochemistry. This is important when validating the scRNAseq results.

iv) Clarify information and name of the antibody denoted as AchR is nicotinic receptor β subunit. Does this correspond to Chrnb2? In this case, please state the full name of the antigen. AchR is a blunt denotation. Moreover, it is a rat antibody not a chicken antibody.

v) ensure that information in the figures is accessible, in particular the size of text and labels.

vi) assess whether neural-crest derived fibroblasts are present in your data sets, given that you find Col1a1, Acta, etc.).

Smaller corrections to the text and figures:

1) First sentence in the abstract – (describe also the origin from nerve-associated SCP-cells, and state gastrointestinal tract not "intestinal tract".

2) Second section introduction: "During development, these cells migrate from the neural tube (not "central nervous system" as neural crest by definition is part of neural tube but not central nervous system – it's a denotion of later structures).

3) p 3 para 2 line 4-5: Much of the ENS is derived…… , enter the foregut and migrate (remove from) caudally to populate…

4) p 3 para 2 line 12 – probably better to stick with caudally rather than switching to posteriorly.

5) For clarity in the last sentence of this section, please indicate what Embryonic day stage 17-18 correspond to.

6) Citation 19 should probably be 14 in last section on subtypes markers describing Figure 7.

7) Figures5 and 6 – In these figures the authors use pre.int and post.int rather than the pre- and post-umbilical used in the text and legend. Please clarify?

8) It is stated that neurons and secretomotor neurons are merged to generate Figure 7. As secretomotor neurons are neurons, it is not clear what is meant here?

9) p 5 para 1 line 1-2 – functionally distinct units have distinct functions by definition.

10) p 4 para 1 – The idea of differences between vagal and sacral crest contributions to ENS is inserted in the middle of this paragraph on enteric neurophathies, but whether the genes mentioned are differentially expressed in the two populations is not stated. This should be clarified.

11) p 11 para 2 line 7 – What is meant by intermediate levels of cells?

12) p 13 para 2 line 2 – These are neural clusters, but some of them are not neuronal, eg those that are glial or are precursors.

13) Figure 1 – Are both the vagal and sacral populations post-umbilical? In C it indicates that sacral is post, but that is not indicated for vagal.

*Reviewer #1 (Recommendations for the authors):*

This manuscript is generally well constructed, although I have a number of minor suggestions below. My major concern is that no functional data are included. The gene expression differences among specific enteric neural subtypes from vagal and sacral neural crest provides the authors with a multitude of hypotheses they could test to validate the many inferences they make throughout the manuscript. Testing at least one of these, for example showing whether a gene knockout has more of an effect on a sacral population or a vagal population would significantly enhance the importance of this manuscript for the field. This is especially the case since with the exception of vascular muscle and melanocytes, it seems like vagal and sacral crest can generate similar derivatives, although in very different proportions. So it is important to understand whether or not there is any compensation or regulation when a subset of one population is diminished. In addition, the authors suggest in a number of places that environmental cues influence cell fate decisions. However this remains untested in the context of these new lineage tracing studies.

*Reviewer #2 (Recommendations for the authors):*

Suggestions to improve the manuscript:

Relating to issue 1: Using regular immunohistochemistry – can you validate EGFP expression in the populations that may be arising from labelled cells? Another alternative explanation to the capture of non-ENS cells in the data could be (as you also mention) contamination during FACS, but also potential leakage of the virus when injecting into the neural tube. It is possible that small amounts of virus escape and infects the gut primordia (which is also easily accessible at this stage in the chicken). These alternative explanations should be tested and discussed.

Relating to issue 4: Please provide separate analysis of vagal (pre and postumbilical), and sacral (postumbilical) and analyse the differentiation paths with relevant methodologies (Velocity?) if possible. The UMAP in its current state is hard to interpret. As reviewer I cannot address the claims made on cellular identities without access to lists of differentially expressed genes.

Relating to issue 6: Clusters could be named after the most prominent marker, or only by numbers in figures. It is fine to speculate on which functional groups they may correspond to, but leave those out from the figures, or make sure it is clear that the functional names are presumed/potential/putative. Nomenclature is an alarming complication in the ENS field at the moment, where highly speculative names are given to scRNA-seq clusters – these tend to damage and confuse the field (seen in recent papers including Drokhylansky et al., 2020 and Fawkner-Corbett et al., 2021). For example, these publications include the errorness use of CGRP as sensory marker (it is expressed in at least 4 other functionally distinct ENS classes) and the made-up "neuroendocrine enteric neurons". It would be valuable to keep an open mind in the chicken system and allow the classification system to grow in a reasonable pace based on functional validations. I hope you agree!

Related to issue 7: It would be advisable to use other colour combinations, and larger pictures displaying the cells in question. The zoomed-out gut with DAPI can be shown ones, as they don't add much information.

Related to issue 8: Marker analysis – please provide a more full-bodies analysis to support the claims that neural crest from different axial regions have different fate potencies. As an example, in Figure 6 it would be important to define the proportion of total neurons that expresses the marker genes. Please also define myenteric versus submucosal analysis. Make sure to align your validating IHC to the markers found in the scRNA-seq. In the case of AchR – why is this used? (it correspond to Chrnb2 which is not part of the scRNA-seq analysis). Other relevant markers such as Sst, Npy and more are however not shown. If adrenergic/dopaminergic/serotonergic neurons are claimed, please show IHC for these neurotransmitters (related to earlier comment most of these phenotypes aren't well validated in the mouse, and especially not in the chicken). An important extension of your work would be to analyse the markers in older chickens, this could partially resolve the issue you have in analysing the ENS with scRNA-seq at such an early stage of neuronal development.

Related to issue 9: Corrections on gene-cell-function correlations: Grm3 is the receptor for glutamate, not a sign of glutamate production. It is stated that excitatory motor neurons express Gad1, Calb2 and Nts, while Calretinin is expressed in excitatory motor neurons, there is no evidence that they would be GABAergic in other species. How Nts relates to this phenotype is also not clear. Gfra3 is the receptor for Artemin, not GDNF. Expression of neuroblast marker Hes5 could indicate differential maturity between vagal and sacral neural crest. Please state the paper showing connection between Myo9d and P75. Expression of Gal is not a sign of excitatory MN, rather inhibitory MN. Ntg1 is not a marker of IPAN, Noggin is correlated to IPANs in the mouse, but not a bona fide marker. How Calbindin relates to IPANs in the chicken is not clear, and in mouse it is also not a good marker, it is rather mostly not correlated (Morarach et al., 2021). Fut9 is in Morarach et al., a plausible marker of excitatory MN, not inhibitory MN, although not validated. Discussion: It is mentioned that Cck, Vip, Sst, Nog and Nmu are sensory markers. While Nmu was shown in Morarach et al., to specifically mark murine IPANs, Cck was shown to mark Dogiel Type 1 cells, and not classical IPANs. VIP and SST are excluded from IPANs in the mouse, while SST is expressed in human IPANs.

Additional issues/questions:

1) Some markers found (Col1a1, Acta2 and more) are also associated with neural crest-derived fibroblasts (potential mesothelial cells according to Zeisel et al., 2018 or mesenchymal cells; Ling and Sauka-Spengler, Nat Cell Biol. 2019). Please compare gene expression patterns to explore if you can find the equivalent cells in your dataset.

2) Which is your definition of schwann cells? By definition, most ENS researchers tend to define enteric glia as those part of the ganglionic plexi or scattered within muscles/villi, while schwann cell precursors giving rise to ENS are nerve-attached. Mpz is found in developing ENS marking presumed schwann-cell precursors (Morarach et al., 2021). Perhaps calling them schwann cell precursors would be ok, although if Mpz expression is found within ganglia and not attached to nerves?

3) It is highly advisable to perform scRNA-sequencing at a stage when the ENS is mature, or if not possible, at the very least at a stage when the full plethora of ENS neuron classes have differentiated from both sacral and vagal origins. Only then would you be able to evaluate the difference in fate between the two axial regions of the neural crest.

4) A recommendation would also be to scrutinize the scRNA-seq dataset from the human ENS (Elmentaite et al., Nature 2021). Markers conserved between the mouse (Morarach et al., 2021) and human may be the most relevant to assess in your scRNA-seq chicken dataset.

5) Figures need more attention. Figure 2: A – instead of displaying all genenames (which are unreadable), consider to name examples of genes instead. B – Again, the text is unreadable, please make bigger C- The C0-12 index needs to be larger and please put in the numbering in the UMAP. Comments on Figure 5-6 are already provided above. Figure 7: Images are stretched in general. Provide larger numbering in A, and bigger index.

Smaller corrections:

1) First sentence in the abstract – (describe also the origin from nerve-associated SCP-cells, and state gastrointestinal tract not "intestinal tract".

2) Second section introduction: "During development, these cells migrate from the neural tube (not "central nervous system" as neural crest by definition is part of neural tube but not central nervous system – it's a denotion of later structures).

3) Much of the ENS is derived…… , enter the foregut and migrate (remove from) caudally to populate…

4) For clarity in the last sentence of this section, please indicate what Embryonic day stage 17-18 correspond to.

5) Citation 19 should probably be 14 in last section on subtypes markers describing Figure 7.

6) Preumbilical ENS express TH in Figure 6X – however in the legend it is stated that small intestine does not express TH. What do you mean?

7) It is stated that neurons and secretomotor neurons are merged to generate Figure 7. As secretomotor neurons are neurons, it is not clear what is meant here?

Comment on data, code, reagents:

1) The chicken ENS has not been analysed by scRNA-seq at this stages before. As these datasete would be very helpful for researchers investigating the chicken ENS as well as to compare with ENS in other species, ideally, much more information on gene expression data are needed. For instance, tables listing the DEG would be a great supplement to the paper.

2) The antibody denoted as AchR is nicotinic receptor β subunit 2. Does this correspond to Chrnb2? In this case, please state the full name of the antigen. AchR is a blunt denotation. Moreover, it is a rat antibody not a chicken antibody.

[Editors' note: further revisions were suggested prior to acceptance, as described below.]

Thank you for resubmitting your work entitled "Single-cell profiling coupled with lineage analysis reveals distinct sacral neural crest contributions to the developing enteric nervous system" for further consideration by *eLife*. Your revised article has been evaluated by Kathryn Cheah (Senior Editor) and a Reviewing Editor.

The manuscript has been improved but there are some remaining issues that need to be addressed, as outlined below:

*Reviewer #2 (Recommendations for the authors):*

Several concerns have been dealt with. Outlined below are the remaining or new issues:

1) In light of Figures 4 and 5, I wonder which part of the peripheral nervous system that actually has been included in the single cell RNA-sequencing experiment. The validating figures indicate that most sacral neural crest derived cells are contained within the Nerves of Remak, which is not part of the ENS. Is also Nerves of Remak (and even sacral ganglia) included in the sequencing? In this case the whole comparison is not between different ENS populations, but rather between the vagal-derived hindgut and Nerve of Remak (potentially consisting of both cells that will stay in this region, and that will migrate into the gut) in essence.

2) There are still a lot of assumptions of the functional neuron types based on neurotransmitters/receptors and other signaling pathways that are premature given the lack of basic characterization of the chicken neural composition. None of the below comments are corroborated in literature:

"Differential gene expression analysis revealed intriguing distinctions between vagal and sacral neural crest cells in the post-umbilical gut at the population level. Genes enriched in the sacral population include SST1/SSTR indicating interneuron cell fate, and DBH, TH, DDC, PNMT, and SLC18A2 which are present in catecholaminergic neurons and serotonergic neurons. GRM3 expression indicates that glutamatergic character is more abundant in the sacral population. In addition, we observed up-regulation of GFRA3 which is involved in the GDNF signaling pathway and CXCL12 which is related to signaling during cell migration (Figure 1C, D). Conversely, the vagal post-umbilical population expresses the adrenergic receptor ADRA1B, enzyme GAD1, CALB2, and NTS consistent with excitatory motor neuron fate."

a) Grm3 encodes the metabotrophic glutamate receptor. Hence it is true to say that the sacral population shows a higher ability to respond via Grm3 in the sacral population, however, it doesn't mean that it has a higher "glutamatergic character". Note that glutamate can be sensed by various receptor complexes (NMDA, AMDA, metabotrophic receptors), chances are therefore high that glutamate is sensed by other receptors in the vagal population.

b) SST1 and SSTR – there is no evidence for the linkage to chicken interneurons (please cite if this is the case). Receptors are found in various populations in the mouse and human, and while somatostatin is specific to interneurons in the mouse, somatostatin is expressed both in IPANs and interneurons in the human.

c) Gfra3 is the receptor for artemin. Note also that Gfra3 is associated with SCP in the developing mouse (Morarach et al., 2021)

It continues:

"Due to the expression of the secretomotor neuropeptide VIP and the tachykinin TAC1, we classified C2 as a motor neuron cluster that is predominantly derived from the vagal neural crest and is present in the pre-umbilical and post-umbilical gut (Figure 2C)."

d) VIP is expressed in several interneuronal subtypes in the mouse in addition to the motor neurons, and it is not only in secretomotor neurons, but also in inhibitory motor neurons of the myenteric plexus. Likewise, Tac1 is a very broadly expressed gene, also in interneurons not only motor neurons. In the human Tac1 is expressed in IPANs. Amongst motor neurons in the mouse, VIP is confined to inhibitory motor neurons, while Tac1 is expressed in excitatory motor neurons. Thus, you cannot conclude functional belonging based on these markers. You can speculate, but it would be inappropriate to reiterate this speculation in the abstract and major results.

And furthermore:

"DBH, TH, DDC, PNMT, and SLC18A2 which are present in catecholaminergic neurons and serotonergic neurons"

e) It is important to be aware of the large population of ENS cells in the intestine goes through a transient state where TH and to some extent DBH is expressed. It is thus unclear whether DBH and TH indeed are maintained in this neural population. If Nerve of Remak is also included in the sequencing these cells may represent sympathetic/parasympathetic neurons.

f) In general in order to compare subtypes across species it is not advisable to use the neurotransmitters/peptides only as these often vary substantially. Transcription factors that actually specify cell types are more likely to faithfully demarcate neuron types. As an example, Pbx3 expression is likely to indicate similarity to likely interneuron type (ENC12, Morarach et al., 2021), especially given the strong coexpression of Penk and Nefm. It would be advisable to screen and compare transcription factors if you would like to infer possible functional belongings.

3) The authors have not confirmed that non-ENS cells indeed stem from the labelled cells (question in previous round). Given that the fluorescent is rather low, it would be to expect that contaminating cells are captured during FACS. It is generally not possible to completely isolate a single cell population based on rather weak reporters used from the gut, which contains a large number of auto-flourescent cells. Moreover, consider that the enteric neural progenitors have migrated through the entire gut to reach the colonic region due to their native ability to do so. It sounds unreasonable that also for instance fibroblasts and epithelial cells would take the effort to migrate that far. It is highly uncertain that most of the indicated non-ENS cells actually are stemming from the vagal/sacral neural crest. It should be discussed in the manuscript that non-ENS cells may be contaminating cells, but that it cannot be excluded that some may be true traced cells. If you still believe that they are traced cells, please show pictures indicating co-expression between your reporter and markers of the various non-ENS cells.

4) Related to the question on whether Nerve of the Remak cells are included – if not included, and only ENS cells were captured, it also remains possible that the missing progenitor population resides within the pelvic ganglion. Sampling could lead to whether populations are included or not. Again, it will be important for the message and interpretation of data that readers understand which peripheral nervous system parts are included in the sequencing experiment.

5) As it takes several more days for sacral neural crest to reach the gut, and that may first differentiate somewhat (most likely undertaking a schwann-cell character) within the Nerve of Remak/Pelvic Ganglia, it is not surprising that the progenitor population express different genes in vagal versus sacral gut.

6) If Nerve of Remak is included, it is questionable how relevant the comparison is to ENS of humans and disease mechanisms of Hirschsprungs disease, as suggested in the discussion.

7) Please try to compress the results part. As the same datasets are analysed several times by different means it gets a bit repetitive, especially with regards to the neural markers, which are mentioned first as being expressed together, and then more refined in the second analysis. It is enough to talk about them in the more refined analysis (also in light of the comments above, with less functional attributes).

*Reviewer #3 (Recommendations for the authors):*

Abstract l33-34 'suggesting an important role for environmental factors promoting

differentiation of both vagal and sacral neural crest-derived cells in the post-umbilical gut' I am not quite clear of the emphasis the authors intend here. It is surely the case that local environments play a role at some level in the exact choice of individual cells, but the overlapping fates from vagal and sacral crest might reflect shared environment in gut, or simply overlap in endogenous potential. I wonder if this phrase is necessary – maybe consider deleting?

I consider that a number of speculative conclusions are phrased more strongly than appropriate based on the evidence presented. These include:

L38 'apparently lacks such a cluster' is too strong – cells of cluster 0 are present in sacral cells, just population is proportionately fewer. Suggest replace with 'is depleted in this cluster at this stage'.

And

L335 Can the authors distinguish developmental and timing differences? Could it be that potential is similar, but timing differences (perhaps driven by location differences, given that most sacral cells are found in Nerve of Remak?) explain the relative proportion of cells in the clusters, especially c0? Hence I suggest replacing 'and' with 'and/or'. Similarly, the conclusion in l409-413 may also be explained by the difference in location of most sacral cells. This possible caveat to these interpretations should be stated. Furthermore, the impact of this should be explored in the Discussion, as a caveat to any interpretations of the cell-types identified in each of the profiles.

L160 The authors state 26993 cells score positively for the RIA transcripts, but they should clarify whether all of these fulfil their criterion of >200 genes expressed i.e. what are the total numbers of vagal and sacral NCCs included in analysis? They should also add information on the numbers of genes identified per cell, e.g. what is the mean number of genes identified per single cell. The MandM indicate that cells with only 200 genes detected were included in the analysis (l561-562), but it is important to understand what the mean and variance was. This has implications for distinguishing putative cell-types (clusters) of different potency, as discussed recently by the Kelsh lab (see Kelsh et al., 2022, Development; Subkhankulova et al., 2023 Nature Comms.). For example, Figure 3 shows that C6 and C8, both classified by authors as glial, show low, but detectable levels of MITF, a key melanocyte determinant; this implies these cells retain melanocyte potential, although the authors do not comment on this. However, the extent to which other cells retain melanocyte or other fate potential is highly dependent upon the depth of sequencing (number of genes detected per single cell).

Figure 4 and 5 All colours should be shown independently, as well as merged, to clarify overlaps.

Figure 5 shows that most sacral cells reside in Nerve of Remak. Differences in relative contribution of sacral v vagal crest are clear in Figure 2C, but only absolute differences seem to be in small cluster 12 (?mesenchymal stem cell). To what extent does this bias in location of cells explain the bias in fates adopted? Put another way, if you consider the relative abundance of sacral versus vagal cells in the hindgut and Ganglion of Remak, are the apparent differences in certain cell-types (clusters) explained by sampling error? Likewise, in Figure 6, where the neuronal populations are analysed in more detail, there are some differences, but absolute differences are in relatively minor populations. This makes me hesitant to conclude that there are substantial differences in potential between the two populations, although this seems in places to be the authors' conclusion (e.g. l335, l484-486). The authors should add a discussion of this point, and soften their conclusions accordingly.

Figure 7 Authors should explain the meaning and significance of 'latent time' and 'absorption probabilities in panels B and C respectively.

[Editors' note: further revisions were suggested prior to acceptance, as described below.]

Thank you for resubmitting your work entitled "Single-cell profiling coupled with lineage analysis reveals vagal and sacral neural crest contributions to the developing enteric nervous system" for further consideration by *eLife*. Your revised article has been evaluated by Kathryn Cheah (Senior Editor) and a Reviewing Editor.

The manuscript has been improved but there are some remaining issues that need to be addressed, as outlined below:

*Reviewer #2 (Recommendations for the authors):*

The authors have been responsive to most concerns. However some remaining issues are listed below.

1) Introduction/Abstract: Trajectory analysis suggest that sacral neural crest has a predicted terminal fate of enteric glia.

What are the conclusions drawn from this? It likely means that neurons have already been generated and that progenitors only give rise to glia at E10 (which you allude to in the discussion). However, as it is phrased it sounds either like the conclusion is that all sacral cells eventually become glia (even the neurons), or that the sacral cells only become glia (which is not the case since you see neurons in the sacral population). Both the fact that you don't see the neurogenic population, and that the trajectory analysis points to glia indicate that at E10, sacral neurogenesis has ceased and progenitors that are present mainly give rise to glia. However, it is rather likely that this is the case for the Nerve of Remak, as the majority of cells that you capture from the sacral tracings are located here. An alternative conclusion could be that neurogenesis of sacral-derived ENS within the gut wall may occur later than E10 and have not yet started. A stringent comparison between cells that are only located in the post-umbilical gut wall that are either sacral or vagal in origin is evidently not performed. Please rephrase sections to make the message more clear on the glial destiny of sacral neural crest, as well as the comparison made in the current study being largely performed between NoR and postumbilical vagal NC. NoR may be equivalent to parts of ENS in other species, so the comparison is not irrelevant, but it needs to be more clearly explained so that no misunderstandings are created in the field.

2) On the same matter. That sacral neural crest can give rise to glia cells in NoR is not contrary to any conception in the field as far as this reviewer can tell. Therefore it is not clear why so much emphasis is placed on this observation (for instance "Validation of sacral contribution to glial fates" as its own subsection and Figure). It would be sufficient to conclude that no or only a small neuroblast cluster is captured from sacral-derived cells at E10, which could indicate that neurogenesis has ceased already. Clear sacral-derived glial clusters are presented in Figure 2, and validation could be presented in association with this figure 2. Remember also that an alternative conclusion could be that neurogenesis of sacral-derived ENS within the gut wall may occur later than E10 and have not yet started.

3) Row 262: These results confirm that a large proportion of the ENS in the colon and Nerve of Remak are derived from the sacral neural crest at E10.

Your data suggest that the cells in the NoR is generated from the sacral neural crest, however a very small proportion of the cells in the colon are generated from the sacral neural crest tracing. Please rephrase. Moreover, the sentence suggests that the ENS is located within the Nerve of Remak, is this nomenclature accepted?

4) Figure 7. RNA velocity analysis makes most sense if only cells are included that are clonally related. Thus, it is not clear why both sacral and vagal sources are included in a common analysis. Please perform individual analysis to understand if it is possible to see bridge populations between progenitor and neural populations (as you suggest there is no active neurogenesis) in the sacral-derived dataset.

5) On the issue of non-ENS cells being captured for sequencing, you have responded that while you cannot confidently confirm fluorescent non-ENS cells, the non-ENS cells have RIA transcripts. However for results presented in Figure 7 you state "In order to better identify potential differences in vagal and sacral contributions to neuronal and glial lineages within the E10 pre- and post-umbilical gut, we isolated cells from RIA+ neuronal (C2, C4), glial (C1, C6, C8), and progenitor clusters (C0, C7, C3) for RNA velocity analysis."

Does this means that the selected cells are the only cells with RIA, or you mean that you selected cells that are known to constitute ENS differentiation (progenitors, neurons and glia)? It would be helpful for this issue if you could present the average number of RIA for each cluster. This could also help guiding you in the conclusion whether the non-ENS cells are resulting from mere contamination (if the RIA is low or non-existing) or likely from true lineage-tracing (if the RIA is the same as the progenitor/neurons/glia). Note also that you in the results parts present the non-ENS cells as likely contaminants, while in the discussion it is more firmly concluded that the non-ENS cells are derived from the neural crest. Please be consistent in the message.

6) Discussion: "The enteric nervous system regulates critical gastrointestinal functions including digestion, hormone secretion and immune interactions."

Note that the role of the ENS in hormone secretion is not known in the gut. What you mean is probably luminal secretion from the epithelia. This is a fluid secretion, not hormones (at least this is what there is evidence for today)

7) Discussion: "Studies have suggested a critical role for the vagal neural crest development and disorders"

What does the sentence mean?

8) Figure 8: Glia in the Nerve of Remak is called "enteric glia" – is this correct, or should they be called schwann cells?

9) It would be advisable to review the figure legends. For instance the Figure 2 title does not reflect the contents and the description of Figure 3A is also not very well aligned with the contents.

10) Discussion: "Both species have Pmp22/Frzb/Cdh19+ glial clusters (C1/6/8), but only the embryonic chick glial clusters expressed Plp1 indicating potential species or stage (embryonic versus adult) differences."

It is not clear what study is referred to and if it is true that Plp1 is not expressed in the murine clusters. At least the juvenile scRNA-seq characterisation of enteric glia in mouse (Zeisel et al., 2018) does show massive expression of Plp1.

11) Discussion Row 479: "Compared with these studies done with postnatal and adult tissue, we observed more 479 clusters with progenitor/precursor identity (Figure 2A, Fig7A), which is not unexpected, given that our analysis utilizes the gut from embryonic stages."

Yes, perhaps not worthwhile mentioning.

12) In general the manuscript would win from a more concise writing, please go through and omit unnecessary information and lengthy description of clusters.

*Reviewer #3 (Recommendations for the authors):*

The authors have addressed in full all points I raised. In particular, the Discussion is now much-improved.

However, I think there is one place where the discussion still over-reaches slightly. At l427 the authors state 'RNA velocity analysis demonstrates that the vagal neural crest maintains a glial/neuronal bipotent progenitor pool while the sacral neural crest does not at E10. This may indicate potential differences in developmental potential, timing, and/or proliferative capacity.'

I think the description of Cluster C0 as a 'glial/neuronal bipotent progenitor pool' is flawed because the RNA velocity analysis only assessed (l308-9) 'neuronal (C2, C4), 309 glial (C1, C6, C8), and progenitor clusters (C0, C7, C3)'; thus, the analysis was restricted to identifying links to neuronal and glia fates only. Furthermore, the depth of sequencing (average <3000 genes/cell) is insufficient to show anything other than strong biases in differentiation state. Hence I would remove the word 'bipotent' which implies that the cells are partially-restricted, when that is simply not assessable from the data provided.

I think there is one key way in which the paper can be made more accessible to the reader. The Figureswith cluster diagrams are difficult for the reader to interpret due to a lack of a key to the clusters, giving their identifying number and their interpretation. A simple key repeated in each figure would solve this.

I noticed a couple of residual typos, e.g. l283 'interneruons', which should be carefully sort and corrected in Word.

In Discussion, I don't think this sentence makes sense: 'Studies have suggested a critical role for the vagal neural crest development and disorders11,18.'

---

## [Author Response]

Essential revisions:1) Extend the strategy of the scRNA study to stages when ENS neurons from both vagal and sacral origin are fully differentiated, compared to the current data set in which the expression profiles indicate that the differentiation states of both populations do not match, including relatively immature cell states. This will allow solid conclusions about cell types contributed by either source of neural crest, as well statements of relative proportions of the different populations.

We appreciate this comment and think it is an excellent suggestion that we definitely plan to do. This made us realize that we failed to clarify in the text why we chose this particular time point for our study, which is two-fold.

First, we are particularly interested in how neural crest cells choose their prospective fates. E10 (similar to E16 mouse and 8wk post-conception in human) is a time when the post-umbilical chick gut has been completely populated by both vagal and sacral neural crest cells for 2 days so cells are in the process of differentiation but not all cells are fully differentiated. For this reason, we can capture both precursors and some differentiated neuronal subtypes. We have clarified this point in the revised manuscript and now focus much more on the precursor population to identify both genes that are common to vagal and sacral neural crest cells as well as those that are distinct. This enables us to formulate testable hypotheses for the potential role of particular transcription factors is allocation of cell fate.

Second, in the US, chick embryos are not considered vertebrates until after E10. Thus, E10 represents the last timepoint we can raise embryos without animal approvals which are not currently in hand. We completely agree that performing experiments at later timepoints will be incredibly valuable and therefore are now applying for approvals. But realistically, these take considerable for approval and thus would delay publication of our datasets for at least another year. Therefore, we propose to publish the mature dataset as a Research Advance that would focus on differences between neuronal subtypes between pre-umbilical vagal, post-umbilical vagal and sacral datasets at more mature timepoints that would nicely complement the current work.

Rather than an additional time points, we have dived deeper into our analysis and discovered an interesting transition in the precursor population. We find that at E10, almost all the sacral precursors are glial in nature whereas the vagal populations (both pre- and post-umbilical) contain a bipotent progenitor pool that can give rise to neuronal or glial fates. Importantly, this suggests that the sacral crest cells are well on their way to differentiating whereas the vagal populations are continuing to maintain a more stem cell-like precursor pool. Second, we see a primary contribution of the vagal preumbilical population to motor neurons while sacral predominantly gives rise to cells with an IPAN/adrenergic signature. Third, this refocusing seems particularly important given that our original aim was to explore differences between vagal and sacral neural crest contributions to the gut. However, the single cell data reveals extensive overlap between sacral and vagal neural crest contributions to the post-umbilical gut, suggesting a strong environmental influence on cell fate decisions.

2) Validate the strategy for neural crest lineage tracing using the RIA retroviruses, given the substantial contribution of non-ENS cell types, such as hematopoietic cells, muscle, endothelial cells, etc.. This would be necessary to support the conclusion that the neural crest contributes to various gut cell types, which is potentially interesting given that multiple corresponding neural crest tracing studies in mouse label only limited or none of these cell populations.Related to this point, it is important to show how far the injected virus spreads, given the viral labelling technique is new for avians.

Based on this question, we have reanalyzed our data including a quality control step that ensured only cells with successful infection were used for downstream analysis. The hematopoetic cells were eliminated in this way. However, neural crest contributions to muscle and melanocytes remained. Regarding validation that the RIA virus is limited to neural tube derived tissue, we are quite skilled at restricting our injections to the lumen of the neural tube given many years of experience performing these injections with many lineage labels. Using RIAs, we have already published four papers using this methodology, many of which are referred to in the manuscript (Li et al., 2019; Tang et al., 2019; 2019; 2021). In these papers, we clearly show that the virus only labels neural tube cells.

However, this was not previously validated for sacral injections. Unfortunately, it is not feasible to perform such an experiment with RIA retroviruses because it requires 48+ hours for integration and expression of the fluorescent protein. Therefore, to demonstrate specificity of our injections, we now include images showing sacral injection using an alternative lineage approach by injecting the vital lipophilic dye DiI into the sacral neural tube in exactly the same manner as we use for performing RIA injections. Embryos were fixed 1 hour or 48 hours post injection. At 1 hour, the DiI is confined to the lumen of the sacral neural tube. By 48 hrs, migrating neural crest cells are clearly visible migrating away from the labeled neural tube. No other tissue is labeled. These data are displayed in Supplemental Figure 1. We thank the reviewers for pointing out that this was not clear in the previous version of the manuscript.

3) Extending the bio-informatic analyses will be necessary to clarify cell type identities as well as the contribution of specific cell populations from the sacral and vagal neural crest, and thereby essential for supporting the conclusions of the study. Specifically, the following points should be addressed:a) In addition of the merged datasets, provide separate analyses of the (pre- and postumbilical), and sacral (postumbilical) populations to elucidate the differentiation paths and relations of individual clusters with relevant methodologies (e.g. Velocity).

This is an excellent point. We now provide separate datasets for each condition as well as trajectory analysis using scVelo and CellRank.

b) Comparison to existing ENS data sets from other species, in particular mammals, will likely help to define the various ENS populations in the chicken and place the results and conclusions of the study into the larger context of the ENS field. (recommended).

Thank you for this excellent suggestion. We now use the markers presented in the dataset of Morarach et al. 2021 at E15.5 (which is close to age of our samples) for assigning gene signatures used for cluster identification. This has greatly aided our analysis and helps put our data in the broader context of regarding ENS development.

c) Related, this is the first scRNAseq study of the chicken ENS at this stage and will therefore represent a valuable resource to the ENS field across all species. Including for instance tables listing the DEG would be a highly useful supplement to the paper.

Thank you for the suggestion, which has been added as a supplement.

4) Validate the scRNAseq data and resulting conclusion by providing more extensive analyses of the different ENS populations employing immunohistochemistry similar to Figures 5 and 6, to i) define neuronal subpopulations and ii) quantify the size of the respective populations, which will support the comparison of the contributions from vagal and sacral neural neural crest and the conclusion that cells originating from different axial origin contribute different fates.

We appreciate this suggestion. Accordingly we now include further evidence using numerous antibody markers for glial cells (Plp1, P0, and GFAP), in addition to neuronal and neurotransmitter markers.

Central to this validation is to consider naming clusters systematically after prominent marker genes rather than possible function, since the physiology of the chicken ENS is currently not well characterised. This will allow for better comparison with other species.[for detailed suggestions see comments by reviewer2]

Thank you for this suggestion. We have now renamed the clusters after prominent markers and agree that this is much more accurate. Indeed we previously struggled with appropriate names for these clusters so very much appreciate this input.

5) Depending on the outcome of the above points, it will be important for the authors to choose an appropriate hypothesis to address functionally to corroborate the implication(s) of the promising scRNAseq results.

Thank you for this suggestion. Our data suggest that there are differences in the neuronal precursor population between vagal and sacral neural crest cells. The new analysis demonstrates that the sacral neural crest cells contribute extensively to enteric glia. This leads us to hypothesize that this may be the reason that sacral neural crest cells appear unable to compensate for ablated vagal neural crest cells either in the intact animal or after transplantation in place of the vagal crest as shown by Burns and Le Douarin (2001). We now include a discussion of this point and recent data that has found aberrant enteric glia in Hirschsprung’s patients.

Secondly, we have included tables of putative genes that may be involved in receiving environmental signals as well as transcription factors within the glial clusters (C1, C6, C8). Thus, this Resource Paper provides an ample list of candidate factors for future functional experiments. However, we believe that the functional testing would take at least a year and is beyond scope of this Resource paper.

Additional points to be addressed:i) Clarify whether there are true melanocytes in the GI-tract preparations or cell populations with a largely similar expression profile.

We observed expression of *Mlana*, *Dct,* and *Mitf* (*cmi9)* in our cluster and indeed see these genes in all of our neural crest datasets, consistent with the possibility that there are either neural crestderived melanocytes or progenitors present in the gut. These marker genes have been used by many other authors to indicate the presence of melanocytes including Chen et al., 2021 who profiled neural crest derived cells in the developing heart.

ii) Discuss the present results from chicken in the context of the work, with respect to neural crest-derived cell attaching to nerve bundles and obtaining a schwann cell precursor-like identity before migrating into the gut, from Uesaka et al. 2015 and Espinosa-Medina et al. 2017.

Thank you for pointing our attention to these very informative papers. We have included a discussion of our data in light of these findings.

iii) use distinguishable colours when representing more than one antibody staining in data panels showing immunohistochemistry. This is important when validating the scRNAseq results.

Thank you for pointing out this issue. Figures 4, 5 and 7 have been redone with better use of colors.

iv) Clarify information and name of the antibody denoted as AchR is nicotinic receptor β subunit. Does this correspond to Chrnb2? In this case, please state the full name of the antigen. AchR is a blunt denotation. Moreover, it is a rat antibody not a chicken antibody.

We appreciate this suggestion and have specified the full name of the antigen as *Chrnb2* in the figures and Chrnb2 (Neuronal nicotinic acetylcholine receptor subunit β-2) as indicated in its reference. In the Materials and methods sections, we have modified the antibody as “rat anti chicken Neuronal nicotinic acetylcholine receptor subunit β-2” for clarification.

v) Ensure that information in the figures is accessible, in particular the size of text and labels.

Done.

vi) Assess whether neural-crest derived fibroblasts are present in your data sets, given that you find Col1a1, Acta, etc.).

Yes thank you for noting this. We do indeed see fibroblasts and have clarified this in the text.

Smaller corrections to the text and figures:1) First sentence in the abstract – (describe also the origin from nerve-associated SCP-cells, and state gastrointestinal tract not "intestinal tract".2) Second section introduction: "During development, these cells migrate from the neural tube (not "central nervous system" as neural crest by definition is part of neural tube but not central nervous system – it's a denotion of later structures).3) p 3 para 2 line 4-5: Much of the ENS is derived…… , enter the foregut and migrate (remove from) caudally to populate…4) p 3 para 2 line 12 – probably better to stick with caudally rather than switching to posteriorly.5) For clarity in the last sentence of this section, please indicate what Embryonic day stage 17-18 correspond to.6) Citation 19 should probably be 14 in last section on subtypes markers describing Figure 7.7) Figures5 and 6 – In these figures the authors use pre.int and post.int rather than the pre- and post-umbilical used in the text and legend. Please clarify?8) It is stated that neurons and secretomotor neurons are merged to generate Figure 7. As secretomotor neurons are neurons, it is not clear what is meant here?9) p 5 para 1 line 1-2 – functionally distinct units have distinct functions by definition.10) p 4 para 1 – The idea of differences between vagal and sacral crest contributions to ENS is inserted in the middle of this paragraph on enteric neurophathies, but whether the genes mentioned are differentially expressed in the two populations is not stated. This should be clarified.11) p 11 para 2 line 7 – What is meant by intermediate levels of cells?12) p 13 para 2 line 2 – These are neural clusters, but some of them are not neuronal, eg those that are glial or are precursors.13) Figure 1 – Are both the vagal and sacral populations post-umbilical? In C it indicates that sacral is post, but that is not indicated for vagal.

Thank you for these suggestions changes all of which have been made.

Reviewer #1 (Recommendations for the authors):This manuscript is generally well constructed, although I have a number of minor suggestions below. My major concern is that no functional data are included. The gene expression differences among specific enteric neural subtypes from vagal and sacral neural crest provides the authors with a multitude of hypotheses they could test to validate the many inferences they make throughout the manuscript. Testing at least one of these, for example showing whether a gene knockout has more of an effect on a sacral population or a vagal population would significantly enhance the importance of this manuscript for the field. This is especially the case since with the exception of vascular muscle and melanocytes, it seems like vagal and sacral crest can generate similar derivatives, although in very different proportions. So it is important to understand whether or not there is any compensation or regulation when a subset of one population is diminished. In addition, the authors suggest in a number of places that environmental cues influence cell fate decisions. However this remains untested in the context of these new lineage tracing studies.

We appreciate the reviewer’s suggestion to include functional experiments and we hope to conduct these in the future. Our main goal here was to generate a Resource that provides insights into possible driving factors that may influence differentiation and to establish a list of candidate factors for future testing. Performing long-term knockouts in the chick system requires CRISPR electroporation coupled with neural tube transplantation (as we recently published in Gandhi et al., 2021) which is quite an arduous and time-consuming experiment.

Reviewer #2 (Recommendations for the authors):Suggestions to improve the manuscript:Using regular immunohistochemistry – can you validate EGFP expression in the populations that may be arising from labelled cells? Another alternative explanation to the capture of non-ENS cells in the data could be (as you also mention) contamination during FACS, but also potential leakage of the virus when injecting into the neural tube. It is possible that small amounts of virus escape and infects the gut primordia (which is also easily accessible at this stage in the chicken). These alternative explanations should be tested and discussed.

Yes, we have done this as shown in Figure 5 and 6.

Please provide separate analysis of vagal (pre and postumbilical), and sacral (postumbilical) and analyse the differentiation paths with relevant methodologies (Velocity?) if possible. The UMAP in its current state is hard to interpret. As reviewer I cannot address the claims made on cellular identities without access to lists of differentially expressed genes.

Excellent suggestion. This is now included.

Clusters could be named after the most prominent marker, or only by numbers in figures. It is fine to speculate on which functional groups they may correspond to, but leave those out from the figures, or make sure it is clear that the functional names are presumed/potential/putative. Nomenclature is an alarming complication in the ENS field at the moment, where highly speculative names are given to scRNA-seq clusters – these tend to damage and confuse the field (seen in recent papers including Drokhylansky et al., 2020 and Fawkner-Corbett et al., 2021). For example, these publications include the errorness use of CGRP as sensory marker (it is expressed in at least 4 other functionally distinct ENS classes) and the made-up "neuroendocrine enteric neurons". It would be valuable to keep an open mind in the chicken system and allow the classification system to grow in a reasonable pace based on functional validations. I hope you agree!

Excellent suggestion. We have changed the nomenclature accordingly.

It would be advisable to use other colour combinations, and larger pictures displaying the cells in question. The zoomed-out gut with DAPI can be shown ones, as they don't add much information.

Excellent suggestion. We have changed the figures accordingly.

Related to issue 8: Marker analysis – please provide a more full-bodies analysis to support the claims that neural crest from different axial regions have different fate potencies. As an example, in Figure 6 it would be important to define the proportion of total neurons that expresses the marker genes. Please also define myenteric versus submucosal analysis. Make sure to align your validating IHC to the markers found in the scRNA-seq. In the case of AchR – why is this used? (it correspond to Chrnb2 which is not part of the scRNA-seq analysis). Other relevant markers such as Sst, Npy and more are however not shown. If adrenergic/dopaminergic/serotonergic neurons are claimed, please show IHC for these neurotransmitters (related to earlier comment most of these phenotypes aren't well validated in the mouse, and especially not in the chicken). An important extension of your work would be to analyse the markers in older chickens, this could partially resolve the issue you have in analysing the ENS with scRNA-seq at such an early stage of neuronal development.

We appreciate the helpful suggestions. Because we are using RIA labeling, we can only validate vagal vs sacral cells using antibodies to double-label individual cells and therefore must rely on available antibodies that work in the chick. Thus, we used Chrnb2 Ab according to its availability and that it would likely cross-react with Chrna5 and Chrnb4 which were abundantly expressed in the RIA labelled cells (C4, C5). We used TH and DBH as markers for adrenergic/dopaminergic/serotonergic neurons. We agree with the reviewer and examined Sst, but we could not obtain a reliable antibody for use in chicken.

Given the size of the chick gut and the fact that RIA does not stain the entire population, we are unable to accurately quantitate from our viral injection data. Thus, we have selected representative sections for illustrative purposes. Quantitative analysis can only be done from the single cell data since a few cross sections are not representative of the entire gut length and not all neural crest derived cells are labelled with the RIA.

Corrections on gene-cell-function correlations: Grm3 is the receptor for glutamate, not a sign of glutamate production. It is stated that excitatory motor neurons express Gad1, Calb2 and Nts, while Calretinin is expressed in excitatory motor neurons, there is no evidence that they would be GABAergic in other species. How Nts relates to this phenotype is also not clear. Gfra3 is the receptor for Artemin, not GDNF. Expression of neuroblast marker Hes5 could indicate differential maturity between vagal and sacral neural crest. Please state the paper showing connection between Myo9d and P75. Expression of Gal is not a sign of excitatory MN, rather inhibitory MN. Ntg1 is not a marker of IPAN, Noggin is correlated to IPANs in the mouse, but not a bona fide marker. How Calbindin relates to IPANs in the chicken is not clear, and in mouse it is also not a good marker, it is rather mostly not correlated (Morarach et al., 2021). Fut9 is in Morarach et al., a plausible marker of excitatory MN, not inhibitory MN, although not validated. Discussion: It is mentioned that Cck, Vip, Sst, Nog and Nmu are sensory markers. While Nmu was shown in Morarach et al., to specifically mark murine IPANs, Cck was shown to mark Dogiel Type 1 cells, and not classical IPANs. VIP and SST are excluded from IPANs in the mouse, while SST is expressed in human IPANs.

Thank you for this very helpful information. We have now used this information to assign our clusters and are extremely grateful to the reviewer for this comment, which has helped us to restructure our analysis. We heavily consulted the ENS single cell atlases published by Morarach et al. (mouse) and Drokhlyansky et al. (human) to postulate neuronal subtypes.

We have since re-analyzed the gene lists for all clusters and have included citations for gene markers. Upon this re-analysis, we have found better identification genes for our progenitor and stem cell clusters that are not *Myo9d* and *P75.*

Additional issues/questions:1) Some markers found (Col1a1, Acta2 and more) are also associated with neural crest-derived fibroblasts (potential mesothelial cells according to Zeisel et al., 2018 or mesenchymal cells; Ling and Sauka-Spengler, Nat Cell Biol. 2019). Please compare gene expression patterns to explore if you can find the equivalent cells in your dataset.

Good point. We now have identified a neural crest-derived fibroblast cluster based on these previously published fibroblast markers.

2) Which is your definition of schwann cells? By definition, most ENS researchers tend to define enteric glia as those part of the ganglionic plexi or scattered within muscles/villi, while schwann cell precursors giving rise to ENS are nerve-attached. Mpz is found in developing ENS marking presumed schwann-cell precursors (Morarach et al., 2021). Perhaps calling them schwann cell precursors would be ok, although if Mpz expression is found within ganglia and not attached to nerves?

We agree that we have clusters that could either be called Schwann cell precursors or enteric glia given that there is heterogeneity in these populations and we identify three clusters that are characterized by these markers. We now call these Enteric Glia 1, 2 and 3 and posit that they may be differentially localized but cannot really parse them based on the single cell data.

3) It is highly advisable to perform scRNA-sequencing at a stage when the ENS is mature, or if not possible, at the very least at a stage when the full plethora of ENS neuron classes have differentiated from both sacral and vagal origins. Only then would you be able to evaluate the difference in fate between the two axial regions of the neural crest.

We agree and hope to do this at a later time but it is beyond the scope of our current study.

4) A recommendation would also be to scrutinize the scRNA-seq dataset from the human ENS (Elmentaite et al., Nature 2021). Markers conserved between the mouse (Morarach et al., 2021) and human may be the most relevant to assess in your scRNA-seq chicken dataset.

Thank you for this suggestion. We now link the markers we use to those in mouse and chick.

5) Figures need more attention. Figure 2: A – instead of displaying all genenames (which are unreadable), consider to name examples of genes instead. B – Again, the text is unreadable, please make bigger C- The C0-12 index needs to be larger and please put in the numbering in the UMAP. Comments on Figure 5-6 are already provided above. Figure 7: Images are stretched in general. Provide larger numbering in A, and bigger index.

Thank you for bringing this to our attention. We have now improved the figures.

[Editors’ note: what follows is the authors’ response to the second round of review.]

The manuscript has been improved but there are some remaining issues that need to be addressed, as outlined below:Reviewer #2 (Recommendations for the authors):Several concerns have been dealt with. Outlined below are the remaining or new issues:1) In light of Figures 4 and 5, I wonder which part of the peripheral nervous system that actually has been included in the single cell RNA-sequencing experiment. The validating figures indicate that most sacral neural crest derived cells are contained within the Nerves of Remak, which is not part of the ENS. Is also Nerves of Remak (and even sacral ganglia) included in the sequencing? In this case the whole comparison is not between different ENS populations, but rather between the vagal-derived hindgut and Nerve of Remak (potentially consisting of both cells that will stay in this region, and that will migrate into the gut) in essence.

We are grateful that the reviewer brought this to our attention as this raises an important point. The reviewer rightly points out that the Nerve of Remak is closely associated with the hindgut in the chick embryo. It is the staging ground for many sacral neural crest-derived cells to migrate to the gut. According to Burns and LeDouarin, sacral neural crest cells from the Nerve of Remak and pelvic plexus migrate into the hindgut along extrinsic axons and colonize the hindgut in large number by E10. Therefore, we did include both the Nerve of Remak and the caudal part of the gut which contains the pelvic plexus in our dissections.

We now clarify the dissection process in the Materials and methods and have amended the Figure description to make this more clear. We also include an image of the dissected gut for clarification in supplemental figure 2.

2) There are still a lot of assumptions of the functional neuron types based on neurotransmitters/receptors and other signaling pathways that are premature given the lack of basic characterization of the chicken neural composition. None of the below comments are corroborated in literature:

Thank you for pointing out the over-interpretations below. Accordingly, we have tried to remove all speculation and changed the sentences as described below:

"Differential gene expression analysis revealed intriguing distinctions between vagal and sacral neural crest cells in the post-umbilical gut at the population level. Genes enriched in the sacral population include SST1/SSTR indicating interneuron cell fate, and DBH, TH, DDC, PNMT, and SLC18A2 which are present in catecholaminergic neurons and serotonergic neurons. GRM3 expression indicates that glutamatergic character is more abundant in the sacral population. In addition, we observed up-regulation of GFRA3 which is involved in the GDNF signaling pathway and CXCL12 which is related to signaling during cell migration (Figure 1C, D). Conversely, the vagal post-umbilical population expresses the adrenergic receptor ADRA1B, enzyme GAD1, CALB2, and NTS consistent with excitatory motor neuron fate."

Changed to: "Differential gene expression analysis revealed intriguing distinctions between vagal and sacral neural crest cells in the post-umbilical gut at the population level. Genes enriched in the sacral population include SST1/SSTR, DBH, TH, DDC, PNMT, and SLC18A2. GRM3 expression is more abundant in the sacral population. In addition, we observed up-regulation of GFRA3, the receptor for artemin, and CXCL12 which is related to signaling during cell migration (Figure 1C, D).

Conversely, the vagal post-umbilical population expresses the adrenergic receptor ADRA1B, enzyme GAD1, CALB2, and NTS."

"Due to the expression of the secretomotor neuropeptide VIP and the tachykinin TAC1, we classified C2 as a motor neuron cluster that is predominantly derived from the vagal neural crest and is present in the pre-umbilical and post-umbilical gut (Figure 2C)."

Changed to: "C2 expresses the neuropeptide VIP and tachykinin (TAC1) and is predominantly vagal-derived with 81% of cells contributed from the post-umbilical population and 16% from the pre-umbilical."

"DBH, TH, DDC, PNMT, and SLC18A2 which are present in catecholaminergic neurons and serotonergic neurons"

Changed to: "Genes enriched in the sacral population include *SST1/SSTR, DBH, TH, DDC, PNMT*, and *SLC18A2. GRM3* expression is more abundant in the sacral population, which may reflect a transient state of differentiation."

e) It is important to be aware of the large population of ENS cells in the intestine goes through a transient state where TH and to some extent DBH is expressed. It is thus unclear whether DBH and TH indeed are maintained in this neural population. If Nerve of Remak is also included in the sequencing these cells may represent sympathetic/parasympathetic neurons.

Thank you for this suggestion. We have now gone back over our data to associate clusters with transcription factors in addition to neurotransmitters (Supplemental Table 4). Of note, there is not much literature regarding the role of TFs in particular lineage allocation in the developing ENS with the exception of Pbx3

3) The authors have not confirmed that non-ENS cells indeed stem from the labelled cells (question in previous round). Given that the fluorescent is rather low, it would be to expect that contaminating cells are captured during FACS. It is generally not possible to completely isolate a single cell population based on rather weak reporters used from the gut, which contains a large number of auto-flourescent cells. Moreover, consider that the enteric neural progenitors have migrated through the entire gut to reach the colonic region due to their native ability to do so. It sounds unreasonable that also for instance fibroblasts and epithelial cells would take the effort to migrate that far. It is highly uncertain that most of the indicated non-ENS cells actually are stemming from the vagal/sacral neural crest. It should be discussed in the manuscript that non-ENS cells may be contaminating cells, but that it cannot be excluded that some may be true traced cells. If you still believe that they are traced cells, please show pictures indicating co-expression between your reporter and markers of the various non-ENS cells.

Point taken. In the revised version, we have erred on the side of caution as the reviewer suggests regarding the non-ENS cells, which indeed represent relatively minor populations. We tried to verify these with antibody markers but were unable to find antibodies (e.g. Mitf) that cross-reacts for immunohistochemistry in the chick. Therefore, we now state that although the non-ENS cell clusters were identified based on detectable RIA transcript levels, we cannot rule out the possibility that some may be contaminating cells captured due to autofluorescence.

4) Related to the question on whether Nerve of the Remak cells are included – if not included, and only ENS cells were captured, it also remains possible that the missing progenitor population resides within the pelvic ganglion. Sampling could lead to whether populations are included or not. Again, it will be important for the message and interpretation of data that readers understand which peripheral nervous system parts are included in the sequencing experiment.

We agree and it may be very interesting in the future to attempt to separately isolate and sequence cells from the Nerve of Remak and the Pelvic ganglion. We now clarify our dissection procedure which includes both of the Nerve of Remak and the caudal part of the gut.

5) As it takes several more days for sacral neural crest to reach the gut, and that may first differentiate somewhat (most likely undertaking a schwann-cell character) within the Nerve of Remak/Pelvic Ganglia, it is not surprising that the progenitor population express different genes in vagal versus sacral gut.

Perhaps but it’s interesting that the vagal pre- and vagal post- populations are quite different whereas the sacral and vagal post-umbilical populations resemble each other quite closely. This speaks to the importance of the local environment in differentiation of both populations of neural crest-derived cells, which is a point we now try to emphasize.

6) If Nerve of Remak is included, it is questionable how relevant the comparison is to ENS of humans and disease mechanisms of Hirschsprungs disease, as suggested in the discussion.

We have removed this discussion point.

7) Please try to compress the results part. As the same datasets are analysed several times by different means it gets a bit repetitive, especially with regards to the neural markers, which are mentioned first as being expressed together, and then more refined in the second analysis. It is enough to talk about them in the more refined analysis (also in light of the comments above, with less functional attributes).

Point taken. The results have been shortened accordingly.

Reviewer #3 (Recommendations for the authors):Abstract l33-34 'suggesting an important role for environmental factors promotingdifferentiation of both vagal and sacral neural crest-derived cells in the post-umbilical gut' I am not quite clear of the emphasis the authors intend here. It is surely the case that local environments play a role at some level in the exact choice of individual cells, but the overlapping fates from vagal and sacral crest might reflect shared environment in gut, or simply overlap in endogenous potential. I wonder if this phrase is necessary – maybe consider deleting?

The reviewer raises a good point and we have amended the abstract to be more circumspect.

I consider that a number of speculative conclusions are phrased more strongly than appropriate based on the evidence presented. These include:L38 'apparently lacks such a cluster' is too strong – cells of cluster 0 are present in sacral cells, just population is proportionately fewer. Suggest replace with 'is depleted in this cluster at this stage'.

Changed to: “cells of cluster 0 are depleted in the sacral population at this stage'

AndL335 Can the authors distinguish developmental and timing differences? Could it be that potential is similar, but timing differences (perhaps driven by location differences, given that most sacral cells are found in Nerve of Remak?) explain the relative proportion of cells in the clusters, especially c0? Hence I suggest replacing 'and' with 'and/or'. Similarly, the conclusion in l409-413 may also be explained by the difference in location of most sacral cells. This possible caveat to these interpretations should be stated. Furthermore, the impact of this should be explored in the Discussion, as a caveat to any interpretations of the cell-types identified in each of the profiles.

This is an excellent point. We now add discussion of this possibility as suggested and are more circumspect about the differences in the progenitor population.

L160 The authors state 26993 cells score positively for the RIA transcripts, but they should clarify whether all of these fulfil their criterion of >200 genes expressed i.e. what are the total numbers of vagal and sacral NCCs included in analysis? They should also add information on the numbers of genes identified per cell, e.g. what is the mean number of genes identified per single cell. The MandM indicate that cells with only 200 genes detected were included in the analysis (l561-562), but it is important to understand what the mean and variance was. This has implications for distinguishing putative cell-types (clusters) of different potency, as discussed recently by the Kelsh lab (see Kelsh et al., 2022, Development; Subkhankulova et al., 2023 Nature Comms.). For example, Figure 3 shows that C6 and C8, both classified by authors as glial, show low, but detectable levels of MITF, a key melanocyte determinant; this implies these cells retain melanocyte potential, although the authors do not comment on this. However, the extent to which other cells retain melanocyte or other fate potential is highly dependent upon the depth of sequencing (number of genes detected per single cell).

Thank you for raising this point of confusion. We have now clarified in the material and methods that the RIA+ analyzed cells fulfill the criterion of >200 genes expressed. We have additionally updated the material and methods to reflect the total number of cells analyzed for each sample, alongside the mean and median of the counts and genes identified per cell.

Figure 4 and 5 All colours should be shown independently, as well as merged, to clarify overlaps.

Thank you for this suggestion which is now included. We have generated supplemental figures showing all channels separately.

Figure 5 shows that most sacral cells reside in Nerve of Remak. Differences in relative contribution of sacral v vagal crest are clear in Figure 2C, but only absolute differences seem to be in small cluster 12 (?mesenchymal stem cell). To what extent does this bias in location of cells explain the bias in fates adopted? Put another way, if you consider the relative abundance of sacral versus vagal cells in the hindgut and Ganglion of Remak, are the apparent differences in certain cell-types (clusters) explained by sampling error? Likewise, in

Thank you for raising this interesting point. To address differences in relative contributions of sacral versus vagal crest-derived cells in various clusters, we now include Supplemental Figures that report the proportion of sacral and vagal-derived cells across the clusters (Supplemental Figure 3) and neuronal subclusters (Supplemental Figure 8).

We have also clarified that the single cell sequencing dataset does include the Nerve of Remak and caudal hindgut which contains the pelvic plexus. Thus we do not believe there is a sampling error that would bias the overrepresentation of one cell type over another based on physical location in the gut.

Figure 6, where the neuronal populations are analysed in more detail, there are some differences, but absolute differences are in relatively minor populations. This makes me hesitant to conclude that there are substantial differences in potential between the two populations, although this seems in places to be the authors' conclusion (e.g. l335, l484-486). The authors should add a discussion of this point, and soften their conclusions accordingly.

We agree and have softened our conclusions regarding difference in developmental potential. It seems that spatial localization may be the most important determining factor for cell fate.

Figure 7 Authors should explain the meaning and significance of 'latent time' and 'absorption probabilities in panels B and C respectively.

Thank you for this suggestion. We have subsequently described in greater detail the meaning and significance of these terms in RNA velocity analysis.

[Editors’ note: what follows is the authors’ response to the third round of review.]

The manuscript has been improved but there are some remaining issues that need to be addressed, as outlined below:Reviewer #2 (Recommendations for the authors):The authors have been responsive to most concerns. However some remaining issues are listed below.1) Introduction/Abstract: Trajectory analysis suggest that sacral neural crest has a predicted terminal fate of enteric glia.What are the conclusions drawn from this? It likely means that neurons have already been generated and that progenitors only give rise to glia at E10 (which you allude to in the discussion). However, as it is phrased it sounds either like the conclusion is that all sacral cells eventually become glia (even the neurons), or that the sacral cells only become glia (which is not the case since you see neurons in the sacral population). Both the fact that you don't see the neurogenic population, and that the trajectory analysis points to glia indicate that at E10, sacral neurogenesis has ceased and progenitors that are present mainly give rise to glia. However, it is rather likely that this is the case for the Nerve of Remak, as the majority of cells that you capture from the sacral tracings are located here. An alternative conclusion could be that neurogenesis of sacral-derived ENS within the gut wall may occur later than E10 and have not yet started. A stringent comparison between cells that are only located in the post-umbilical gut wall that are either sacral or vagal in origin is evidently not performed. Please rephrase sections to make the message more clear on the glial destiny of sacral neural crest, as well as the comparison made in the current study being largely performed between NoR and postumbilical vagal NC. NoR may be equivalent to parts of ENS in other species, so the comparison is not irrelevant, but it needs to be more clearly explained so that no misunderstandings are created in the field.

Thank you for pointing out that this wording is confusing. Accordingly, we have modified the abstract and introduction to clarify our interpretation. We speculate that the sacral progenitors at this time point, many of which reside within the Nerve of Remak, have glial characteristics that may reflect an enteric glia/Schwann cell precursor state; we speculate that these cells can differentiate into either neuronal or glial cell types at later time points. This is consistent with a recent publication (*Nat Commun* 14, 5904 2023) suggesting that enteric glia may reflect a transitional cell state that retains neurogenic potential. However, other sacral neural crest-derived cells at E10 have already differentiated within the gut walls into neurons of the submucosal and myenteric plexuses (Figure 5E), some of which at are ACHR+ cells (Figure 5A”). Given these findings, we agree that it is likely that many sacral-derived enteric glia from the Nerve of Remak may invade the gut wall and differentiate at later time points. These points are further discussed in the introduction and discussion.

2) On the same matter. That sacral neural crest can give rise to glia cells in NoR is not contrary to any conception in the field as far as this reviewer can tell. Therefore it is not clear why so much emphasis is placed on this observation (for instance "Validation of sacral contribution to glial fates" as its own subsection and Figure). It would be sufficient to conclude that no or only a small neuroblast cluster is captured from sacral-derived cells at E10, which could indicate that neurogenesis has ceased already. Clear sacral-derived glial clusters are presented in Figure 2, and validation could be presented in association with this figure 2. Remember also that an alternative conclusion could be that neurogenesis of sacral-derived ENS within the gut wall may occur later than E10 and have not yet started.

Thank you for this comment, with which we agree. There is established literature demonstrating sacral crest contribution to glia of the Nerve of Remak. We have adjusted the text accordingly and included the possibility that the sacral-derived ENS within the gut wall may develop later than E10. Since the glial clusters were so prominent, we included Figure 8 to further validate glial marker expression. However, we would be happy to move Figure 8 as a supplemental figure to the in vivo validation of Figure 2 if the reviewer and/or senior editor would prefer that arrangement.

3) Row 262: These results confirm that a large proportion of the ENS in the colon and Nerve of Remak are derived from the sacral neural crest at E10.Your data suggest that the cells in the NoR is generated from the sacral neural crest, however a very small proportion of the cells in the colon are generated from the sacral neural crest tracing. Please rephrase. Moreover, the sentence suggests that the ENS is located within the Nerve of Remak, is this nomenclature accepted?

Thank you for this comment. Accordingly, we have rephrased it as: “These results confirm that the sacral neural crest contributes both to a large portion of the Nerve of Remak as well as a subset of neurons in the colon at E10.”

4) Figure 7. RNA velocity analysis makes most sense if only cells are included that are clonally related. Thus, it is not clear why both sacral and vagal sources are included in a common analysis. Please perform individual analysis to understand if it is possible to see bridge populations between progenitor and neural populations (as you suggest there is no active neurogenesis) in the sacral-derived dataset.

Thank you for your suggestion. We have removed the combined analysis from Figure 7 and have further highlighted the analysis of individual populations.

5) On the issue of non-ENS cells being captured for sequencing, you have responded that while you cannot confidently confirm fluorescent non-ENS cells, the non-ENS cells have RIA transcripts. However for results presented in Figure 7 you state "In order to better identify potential differences in vagal and sacral contributions to neuronal and glial lineages within the E10 pre- and post-umbilical gut, we isolated cells from RIA+ neuronal (C2, C4), glial (C1, C6, C8), and progenitor clusters (C0, C7, C3) for RNA velocity analysis."Does this means that the selected cells are the only cells with RIA, or you mean that you selected cells that are known to constitute ENS differentiation (progenitors, neurons and glia)? It would be helpful for this issue if you could present the average number of RIA for each cluster. This could also help guiding you in the conclusion whether the non-ENS cells are resulting from mere contamination (if the RIA is low or non-existing) or likely from true lineage-tracing (if the RIA is the same as the progenitor/neurons/glia). Note also that you in the results parts present the non-ENS cells as likely contaminants, while in the discussion it is more firmly concluded that the non-ENS cells are derived from the neural crest. Please be consistent in the message.

Thank you for pointing out that this description was still confusing. We now clarify in the text and methods that Figure 7 was generated only from RIA+ cells (Figure 2). To make this more clear, we have included a table (supplemental table 5) that provides the average number of RIA transcripts for each cluster. Since only cells with a minimum of 2 RIA transcripts were clustered and further analyzed in our study, we believe that these are truly lineage traced neural crest-derived cells. We have adjusted the text of the paper accordingly.

6) Discussion: "The enteric nervous system regulates critical gastrointestinal functions including digestion, hormone secretion and immune interactions."Note that the role of the ENS in hormone secretion is not known in the gut. What you mean is probably luminal secretion from the epithelia. This is a fluid secretion, not hormones (at least this is what there is evidence for today)

Thank you for clarifying. We have adjusted the text accordingly.

7) Discussion: "Studies have suggested a critical role for the vagal neural crest development and disorders"What does the sentence mean?

Apologies that this sentence was unclear. We meant to say that the majority of papers on ENS developmental disorders have focused on the role of the vagal neural crest and did not distinguish between potentially unique contributions from either vagal or sacral neural crest.

8) Figure 8: Glia in the Nerve of Remak is called "enteric glia" – is this correct, or should they be called schwann cells?

Thank you for this comment. Given the markers used to identify these cells, we do not believe we can distinguish between a glial or Schwann cell fate. Therefore we have now refer to them as “enteric glia/Schwann cell precursors”. According to Pachnis and colleagues (*Nat Commun* 14: 5904, 2023) these “enteric glia” may represent a transitional cell state that retains neurogenic potential.

9) It would be advisable to review the figure legends. For instance the Figure 2 title does not reflect the contents and the description of Figure 3A is also not very well aligned with the contents.

Thank you for highlighting this concern. We have updated the figure legends accordingly.

10) Discussion: "Both species have Pmp22/Frzb/Cdh19+ glial clusters (C1/6/8), but only the embryonic chick glial clusters expressed Plp1 indicating potential species or stage (embryonic versus adult) differences."It is not clear what study is referred to and if it is true that Plp1 is not expressed in the murine clusters. At least the juvenile scRNA-seq characterisation of enteric glia in mouse (Zeisel et al., 2018) does show massive expression of Plp1.

Thank you for pointing out this mistake. We have corrected the sentence to reflect the presence of *Plp1+* glia in mice (Drokhlyansky et al., 2020).

11) Discussion Row 479: "Compared with these studies done with postnatal and adult tissue, we observed more 479 clusters with progenitor/precursor identity (Figure 2A, Fig7A), which is not unexpected, given that our analysis utilizes the gut from embryonic stages."Yes, perhaps not worthwhile mentioning.

We appreciate this suggestion and have removed the sentence.

12) In general the manuscript would win from a more concise writing, please go through and omit unnecessary information and lengthy description of clusters.

Thank you for this suggestion. We have condensed the paper accordingly. *Reviewer #3 (Recommendations for the authors):*

The authors have addressed in full all points I raised. In particular, the Discussion is now much-improved.However, I think there is one place where the discussion still over-reaches slightly. At l427 the authors state 'RNA velocity analysis demonstrates that the vagal neural crest maintains a glial/neuronal bipotent progenitor pool while the sacral neural crest does not at E10. This may indicate potential differences in developmental potential, timing, and/or proliferative capacity.'I think the description of Cluster C0 as a 'glial/neuronal bipotent progenitor pool' is flawed because the RNA velocity analysis only assessed (l308-9) 'neuronal (C2, C4), 309 glial (C1, C6, C8), and progenitor clusters (C0, C7, C3)'; thus, the analysis was restricted to identifying links to neuronal and glia fates only. Furthermore, the depth of sequencing (average <3000 genes/cell) is insufficient to show anything other than strong biases in differentiation state. Hence I would remove the word 'bipotent' which implies that the cells are partially-restricted, when that is simply not assessable from the data provided.

Thank you for this suggestion. We have condensed the paper accordingly.

I think there is one key way in which the paper can be made more accessible to the reader. The Figureswith cluster diagrams are difficult for the reader to interpret due to a lack of a key to the clusters, giving their identifying number and their interpretation.

Thank you for the insight. We have updated the cluster diagrams to include the interpretation of each cluster.

A simple key repeated in each figure would solve this.I noticed a couple of residual typos, e.g. l283 'interneruons', which should be carefully sort and corrected in Word.

Thank you for the suggestion. We have corrected the typos.

In Discussion, I don't think this sentence makes sense: 'Studies have suggested a critical role for the vagal neural crest development and disorders11,18.'

Thank for your suggestion. This sentence has been expanded to emphasize the point that a majority of papers on ENS developmental disorders have focused on the role of the vagal neural crest in ENS development or have been unable to parse out the potentially unique contributions of the vagal versus sacral neural crest.